# A2PO: Towards Effective Offline Reinforcement Learning from an Advantage-aware Perspective

**Yunpeng Qing[1], Shunyu Liu[2],[*] Jingyuan Cong[1], Kaixuan Chen[2,3], Yihe Zhou[1], Mingli Song[2,3]**
[1] College of Computer Science and Technology, Zhejiang University
[2] State Key Laboratory of Blockchain and Data Security, Zhejiang University
[3] Hangzhou High-Tech Zone (Binjiang) Institute of Blockchain and Data Security
`{qingyunpeng, liushunyu, kcj51, chenkx, zhouyihe, brooksong}@zju.edu.cn`

## Abstract

Offline reinforcement learning endeavors to leverage offline datasets to craft effective agent policy without online interaction, which imposes proper conservative constraints with the support of behavior policies to tackle the out-of-distribution problem. However, existing works often suffer from the constraint conflict issue when offline datasets are collected from multiple behavior policies, *i.e.,* different behavior policies may exhibit inconsistent actions with distinct returns across the state space. To remedy this issue, recent advantage-weighted methods prioritize samples with high advantage values for agent training while inevitably ignoring the diversity of behavior policy. In this paper, we introduce a novel Advantage-Aware Policy Optimization (A2PO) method to explicitly construct advantage-aware policy constraints for offline learning under mixed-quality datasets. Specifically, A2PO employs a conditional variational auto-encoder to disentangle the action distributions of intertwined behavior policies by modeling the advantage values of all training data as conditional variables. Then the agent can follow such disentangled action distribution constraints to optimize the advantage-aware policy towards high advantage values. Extensive experiments conducted on both the single-quality and mixed-quality datasets of the D4RL benchmark demonstrate that A2PO yields results superior to the counterparts. Our code is available at https://github.com/Plankson/A2PO.

## 1 Introduction

Offline Reinforcement Learning (RL) [11, 4] aims to learn effective control policies from pre-collected datasets without online exploration, and has witnessed its unprecedented success in various real-world applications, including robot control [57, 45, 24, 25], power grid control [2, 49, 26, 36], *etc*. A formidable challenge of offline RL lies in the Out-Of-Distribution (OOD) problem [21], involving the distribution shift between data induced by the learned policy and data collected by the behavior policy. Consequently, the direct application of conventional online RL methods inevitably exhibits extrapolation error [34], where the unseen state-action pairs are erroneously estimated. To tackle this OOD problem, offline RL methods attempt to impose proper conservatism on the learning agent within the distribution of the dataset, such as restricting the learned policy with a regularization term [19, 9] or penalizing the value overestimation of OOD actions [20, 18].

Despite the promising results achieved, offline RL often encounters the constraint conflict issue when dealing with the mixed-quality dataset [40, 12, 1, 28]. Specifically, when training data are collected from multiple behavior policies with distinct returns, existing works still treat each sample constraint

---
[*]Corresponding author

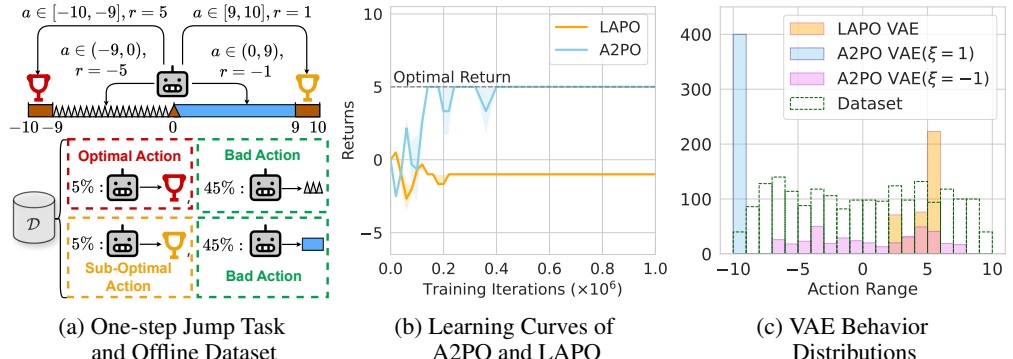

(a) One-step Jump Task and Offline Dataset

(b) Learning Curves of A2PO and LAPO

(c) VAE Behavior Distributions

Figure 1: A didactic experiment. (a) The visualization of the toy one-step jump task and the composition of the mixed-quality dataset. The agent starts at position $0$ and can make a one-step jump $a \in [-10, 10]$ to reach a new position and receive a reward $r$. (b) Learning curves of A2PO and LAPO. (c) VAE-generated action distributions of A2PO and LAPO at the initial state. LAPO VAE conditions only on the state, while A2PO VAE conditions on both the state and the advantage $\xi$.

equally with no regard for the differences in data quality and diversity. This oversight results in improper constrain on conflict actions [5, 16, 50, 15], ultimately leading to further suboptimal outcomes. To resolve this concern, the Advantage-Weighted (AW) methods [5, 43, 58, 28] employ weighted sampling to prioritize training transitions with high advantage values from the offline dataset. However, we argue that these AW methods implicitly reduce the diverse behavior policies associated with the offline dataset into a narrow one from the viewpoint of the dataset redistribution. As a result, this redistribution operation of AW may exclude a significant number of critical transitions during training, imposing erroneous constraints on agent learning. To exemplify the above issue of AW, we conduct a didactic experiment on the recent advanced AW method, LAPO [5], as shown in Figure 1. The toy one-step jump task requires the agent to jump over obstacles and reach two designated goal positions with different rewards. The offline dataset mainly contains failed attempts, with only a few successful transitions, making it very challenging for the agent to learn an effective policy. The results in Figure 1b demonstrate that LAPO performs poorly in this task. Furthermore, Figure 1c reveals that the AW redistribution does not effectively prioritize either optimal or suboptimal actions in modeling the behavior policy. Instead, the AW redistribution can lead to an incorrect on bad actions, which results in unreliable policy optimization.

In this paper, we propose *Advantage-Aware Policy Optimization*, abbreviated as A2PO, to explicitly learn the advantage-aware policy with disentangled behavior policies from the mixed-quality offline dataset. Unlike previous AW methods devoted to dataset redistribution while reducing the data diversity, the proposed A2PO directly conditions the agent policy on the advantage values of all training data without any prior preference. Technically, A2PO comprises two alternating stages, *behavior policy disentangling* and *agent policy optimization*. The former stage introduces a Conditional Variational Auto-Encoder (CVAE) [41] to disentangle different behavior policies into separate action distributions by modeling the advantage values of collected state-action pairs as conditioned variables. The latter stage further imposes an explicit advantage-aware policy constraint on the training agent within the support of disentangled action distributions. The advantage-conditioned CVAE can models the behavior policy distribution (Figure 1c), which is further utilized to construct advantage-aware constraint for agent optimization toward high advantage values, resulting in an effective decision-making policy (Figure 1b).

To sum up, our main contribution is the first dedicated attempt towards advantage-aware policy optimization to alleviate the constraint conflict issue under the mixed-quality offline dataset. The proposed A2PO can achieve advantage-aware policy constraint derived from different behavior policies, where a customized CVAE is employed to infer diverse action distributions associated with the behavior policies by modeling advantage values as conditional variables. Extensive experiments conducted on the D4RL benchmark [8], including both single-quality and mixed-quality datasets, demonstrate that the proposed A2PO method yields significantly superior performance to other advanced offline RL baselines, as well as the advantage-weighted competitors.

## 2 Related Works

**Offline RL** can be broadly classified into four categories: policy constraint [11, 44], value regularization [13, 14], model-based method [55, 52], and return-conditioned supervised learning [7, 23]. Policy constraint methods impose constraints on the learned policy to be close to the behavior policy [19]. Previous studies directly introduce the explicit constraint on policy learning, such as behavior cloning [9], maximum mean discrepancy [19], or maximum likelihood estimation [48]. In contrast, recent efforts [29, 46] mainly focus on realizing the policy constraints implicitly by approximating the formal optimal policy derived from KL-divergence constraint. On the other hand, value regularization methods make constraints on the value function to alleviate the overestimation of OOD action. Researchers try to approximate the lower bound of the value function with the Q-regularization term for conservative action selection [20, 27]. Model-based methods construct the environment dynamics to estimate state-action uncertainty for OOD penalty [17, 6]. Several works also converts offline RL into a return-conditioned supervised learning task. Decision Transformer (DT) [3] builds a transformer policy conditioned on both the current state and the additional sum return signal with supervised learning. Yamagata et al. [51] improve the stitching ability of DT policy on sub-optimal samples by relabeling the return signal with Q-learning results. However, in the context of offline RL with a mixed-quality dataset and no access to the trajectory return signals, all these methods treat each sample equally without considering data quality, thereby resulting in improper regularization and further suboptimal learning outcomes.

**Advantage-weighted Offline RL Method** employs weighted sampling to prioritize training transitions with high advantage values from the offline dataset. To enhance sample efficiency, Peng et al. [31] introduce an advantage-weighted maximum likelihood loss by directly calculating advantage values via trajectory return. [29] further use the critic network to estimate advantage values for advantage-weighted policy training. This technique has been incorporated as a subroutine in other works [18, 50] for agent policy extraction. Recently, AW methods have also been well studied in addressing the constraint conflict issue that arises from the mixed-quality dataset [5, 58, 40]. Several studies present advantage-weighted behavior cloning as a direct objective function [58] or an explicit policy constraint [32]. [5] propose the Latent Advantage-Weighted Policy Optimization (LAPO) framework, which employs an advantage-weighted loss to train CVAE for generating high-advantage actions based on the state condition. Besides AW methods, Hong et al. [15] enhance the classical offline RL training objective with the weight of subsequent return. On the other hand, Hong et al. [16] directly learning the optimal policy density as the weight function to enable sampling from high-performing policies. However, this AW mechanism inevitably diminishes the data diversity in the dataset. In contrast, our A2PO directly conditions the agent policy on both the state and the estimated advantage value, enabling effective utilization of all samples with varying quality.

## 3 Preliminaries

We formalize the RL task as a Markov Decision Process (MDP) [35] defined by a tuple $\mathcal{M} = \langle \mathcal{S}, \mathcal{A}, P, r, \gamma, \rho_0 \rangle$, where $\mathcal{S}$ represents the state space, $\mathcal{A}$ represents the action space, $P : \mathcal{S} \times \mathcal{A} \times \mathcal{S} \to [0, 1]$ denotes the environment dynamics, $r : \mathcal{S} \times \mathcal{A} \to \mathbb{R}$ denotes the reward function, $\gamma \in (0, 1]$ is the discount factor, and $\rho_0$ is the initial state distribution. At each time step $t$, the agent observes the state $s_t \in \mathcal{S}$ and selects an action $a_t \in \mathcal{A}$ according to its policy $\pi$. This action leads to a transition to the next state $s_{t+1}$ based on the dynamics distribution $P$. Additionally, the agent receives a reward signal $r_t$. The goal of RL is to learn an optimal policy $\pi^*$ that maximizes the expected return: $\pi^* = \arg\max_\pi \mathbb{E}_\pi \left[ \sum_{k=0}^\infty \gamma^k r_{t+k} \right]$. In offline RL, the agent can only learn from an offline dataset without online interaction with the environment. In the single-quality settings, the offline dataset $\mathcal{D} = \{(s_t, a_t, r_t, s_{t+1}) \mid t = 1, \cdots, N\}$ with $N$ transitions is collected by only one behavior policy $\pi_\beta$. In the mixed-quality settings, the offline dataset $\mathcal{D} = \bigcup_i \{(s_{i,t}, a_{i,t}, r_{i,t}, s_{i,t+1}) \mid t = 1, \cdots, N\}$ is collected by multiple behavior policies $\{\pi_{\beta_i}\}_{i=1}^M$.

We evaluate the learned policy $\pi$ by the action value function $Q^\pi(s, a) = \mathbb{E}_\pi \left[ \sum_{t=0}^\infty \gamma^t r(s_t, a_t) \mid s_0 = s, a_0 = a \right]$. The state value function is defined as $V^\pi(s) = \mathbb{E}_{a \sim \pi} [Q^\pi(s, a)]$, while the advantage function is defined as $A^\pi(s, a) = Q^\pi(s, a) - V^\pi(s)$. For continuous control, our A2PO implementation uses the TD3 algorithm [10] based on the actor-critic framework as a basic backbone for its robust performance. The actor network $\pi_\omega$, known as the learned policy, is parameterized by $\omega$, while the critic networks consist of the Q-network $Q_\theta$

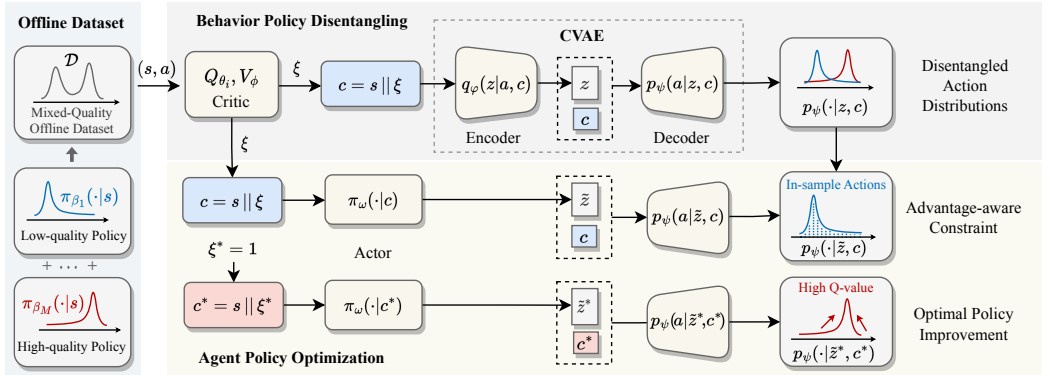

Figure 2: An illustrative diagram of the Advantage-Aware Policy Optimization (A2PO) method.

parameterized by $\theta$ and the V-network $V_\phi$ parameterized by $\phi$. The actor-critic framework involves two steps: policy evaluation and policy improvement. During policy evaluation phase, the Q-network $Q_\theta$ is optimized by the temporal-difference (TD) loss [42]:

$$\mathcal{L}_Q(\theta) = \mathbb{E}_{(s,a,r,s')\sim\mathcal{D}, a'\sim\pi_{\hat{\omega}}(s')}\big[Q_\theta(s,a) - \big(r(s,a) + \gamma Q_{\hat{\theta}}(s',a')\big)\big]^2, \tag{1}$$

where $\hat{\theta}$ and $\hat{\omega}$ are the parameters of the target networks that are regularly updated by online parameters $\theta$ and $\omega$ to maintain learning stability. The V-network $V_\phi$ can also be optimized by the similar TD loss. For policy improvement in continuous control, the actor network $\pi_\omega$ can be optimized by the deterministic policy gradient loss [39, 38]:

$$\mathcal{L}_\pi(\omega) = \mathbb{E}_{s\sim\mathcal{D}}\left[-Q_\theta(s,\pi_\omega(s))\right]. \tag{2}$$

Note that offline RL will impose conservative constraints on the optimization losses to tackle the OOD problem. Moreover, the performance of the final learned policy $\pi_\omega$ highly depends on the quality of the offline dataset $\mathcal{D}$ associated with the behavior policies $\{\pi_{\beta_i}\}$.

## 4 Methodology

In this section, we provide details of our proposed A2PO approach, consisting of two key components: *behavior policy disentangling* and *agent policy optimization*. In the *behavior policy disentangling* phase, we disentangle behavior policies with a CVAE specifically modeling the action distribution conditioned on the advantage values of collected state-action pairs. By taking different advantage inputs, the newly formed CVAE allows the agent to infer distinct action distributions that are associated with various behavior policies. Then in the *agent policy optimization* phase, the action distributions derived from the advantage condition serve as disentangled behavior policies, establishing an advantage-aware policy constraint to guide agent training. An overview of our A2PO is illustrated in Figure 2.

### 4.1 Behavior Policy Disentangling

To realize behavior policy disentangling, we adopt a CVAE to **relate the behavior distribution of different specific behavior policies $\pi_{\beta_i}$ to the advantage condition variables**, which is quite different from previous methods [11, 5, 56] utilizing CVAE only for **approximating the overall mixed-quality behavior policy set $\{\pi_{\beta_i}\}_{i=1}^{M}$ conditioned only on specific state** $s$. Concretely, we have made adjustments to the architecture of the CVAE to be advantage-aware. The encoder $q_\varphi(z|a,c)$ is fed with condition $c$ and action $a$ to project them into a latent representation $z$. Given specific condition $c$ and the encoder output $z$, the decoder $p_\psi(a|z,c)$ captures the correlation between condition $c$ and latent representation $z$ to reconstruct the original action $a$. Unlike previous methods [11, 5, 48] predicting action solely based on the state $s$, we consider both state $s$ and advantage value $\xi$ for CVAE condition. The state-advantage condition $c$ is formulated as:

$$c = s \,\|\, \xi. \tag{3}$$

Therefore, given the current state $s$ and the advantage value $\xi$ as a joint condition, the CVAE model is able to generate corresponding action $a$ with varying quality positively correlated with the advantage value $\xi$. For a state-action pair $(s, a)$, the advantage value $\xi$ can be computed as follows:

$$\xi = \tanh\left(\min_{i=1,2} Q_{\theta_i}(s, a) - V_\phi(s)\right), \tag{4}$$

where two Q-networks with the $\min(\cdot)$ operation are adopted to ensure conservatism in offline RL settings [11]. Moreover, we employ the $\tanh(\cdot)$ function to normalize the advantage condition within the range of $(-1, 1)$. This operation prevents excessive outliers from impacting the performance of CVAE, improving the controllability of generation. The optimization of Q-networks and V-network will be described in the following section.

The CVAE model is trained using the state-advantage condition $c$ and the corresponding action $a$. The training objective involves maximizing the Empirical Lower Bound (ELBO) [41] on the log-likelihood of the sampled minibatch:

$$\mathcal{L}_{\text{CVAE}}(\varphi, \psi) = -\mathbb{E}_{\mathcal{D}}\left[\mathbb{E}_{q_\varphi(z|a,c)}\left[\log(p_\psi(a|z, c))\right] + \alpha \cdot \text{KL}\left[q_\varphi(z|a, c) \,\|\, p(z)\right]\right], \tag{5}$$

where $\alpha$ is the coefficient for trading off the KL-divergence loss term, and $p(z)$ denotes the prior distribution of $z$ setting to be $\mathcal{N}(0, 1)$. The first log-likelihood term encourages the generated action to match the real action as much as possible, while the second KL divergence term aligns the latent variable distribution with the prior distribution $p(z)$.

At each round of CVAE training, a minibatch of state-action pairs $(s, a)$ is sampled from the offline dataset. These pairs are fed to the critic network $Q_\theta$ and $V_\phi$ to get corresponding advantage condition $\xi$ by Eq. (4). Then the advantage-aware CVAE is subsequently optimized by Eq. (5). By incorporating the advantage condition $\xi$, the CVAE captures the relation between $\xi$ and the action distribution of the behavior policies, as shown in the upper part of Figure 2. This further enables the CVAE to generate actions $a$ based on the state-advantage condition $c$ in a manner where the action quality is positively correlated with $\xi$. Furthermore, the advantage-aware CVAE is utilized to establish an advantage-aware policy constraint for agent policy optimization in the next stage.

## 4.2 Agent Policy Optimization

The agent is constructed using the actor-critic framework [42]. The critic comprises two Q-networks $Q_{\theta_{i=1,2}}$ and one V-network $V_\phi$ to approximate the value of the agent policy. The actor, advantage-aware policy $\pi_\omega(\cdot|c)$, with input $c = s \,\|\, \xi$, generates a latent representation $\tilde{z}$ based on the state $s$ and the designated advantage condition $\xi$. This latent representation $\tilde{z}$, along with $c$, is then fed into the decoder $p_\psi$ for an recognizable action $a_\xi$:

$$a_\xi \sim p_\psi(\cdot \mid \tilde{z}, c), \text{ where } \tilde{z} \sim \pi_\omega(\cdot \mid c). \tag{6}$$

With this form, the advantage-aware policy $\pi_\omega$ is expected to produce action with different qualities that are positively correlated to the designated advantage input $\xi$ which is normalized within $(-1, 1)$ in Eq. 4). Therefore, the output optimal action $a^*$ is obtained by $c^* = s \,\|\, \xi^*$ input with $\xi^* = 1$. It should be noted that the critic networks are to approximate the expected value of the optimal policy $\pi_\omega(\cdot|c^*)$. The agent optimization, following the actor-critic framework, encompasses policy evaluation and policy improvement steps. During the policy evaluation step, the critic is updated through the minimization of the temporal difference loss with the optimal policy $\pi_\omega(\cdot|c^*)$. Specifically, for the V-network $v_\phi$, we employ the one-step Bellman operator to approximate the state value under the current agent-aware policy, conditioned on the optimal advantage input $\xi^* = 1$, as follows:

$$\mathcal{L}_{\text{TD}}(\phi) = \mathbb{E}_{\substack{(s,a,r,s') \sim \mathcal{D}, \\ \tilde{z}^* \sim \pi_\omega(\cdot|c^*), \\ a_\xi^* \sim p_\psi(\cdot|\tilde{z}^*, c^*)}} \left[r + \gamma \min_i Q_{\hat{\theta}_i}(s', a_\xi^*) - V_\phi(s)^2\right], \tag{7}$$

where $Q_{\hat{\theta}}$ is the target network updated softly. As for the Q-networks, both of the two Q-network entities $Q_{\theta_i}$ are optimized with agent policy $\pi_\omega(\cdot|c^*)$ following Equation 1.

For the policy improvement, the actor loss is defined as:

$$\mathcal{L}_{\text{AC}}(\omega) = -\lambda \cdot \mathbb{E}_{\substack{s \sim \mathcal{D}, \tilde{z}^* \sim \pi_\omega(\cdot|c^*), \\ a_\xi^* \sim p_\psi(\cdot|\tilde{z}^*, c^*)}} \left[Q_{\theta_1}(s, a_\xi^*)\right] + \mathbb{E}_{\substack{(s,a) \sim \mathcal{D}, \tilde{z} \sim \pi_\omega(\cdot|c), \\ a_\xi \sim p_\psi(\cdot|\tilde{z}, c)}} \left[(a - a_\xi)^2\right], \tag{8}$$

where $a_\xi^*$ in the first term is the optimal action generated with the fixed maximum advantage condition $\xi^* = 1$ input and $a_\xi$ in the second term is obtained with the advantage condition $\xi$ derived from the critic based on Eq. 4 applied to the sampled batch. Meanwhile, following TD3+BC [9], we add a normalization coefficient $\lambda = \alpha / \left( \frac{1}{N} \sum_{(s_i,a_i)} |Q(s_i, a_i)| \right)$ to the first term to keep the scale balance between Q value objective and regularization, where $\alpha$ is a hyperparameter to control the scale of the normalized Q value. The first term encourages the optimal policy condition on $c^*$ to select actions that yield the highest expected returns represented by the Q-value. This aligns with the policy improvement step in conventional RL approaches [22]. The second behavior cloning term explicitly imposes constraints on the advantage-aware policy, ensuring the policy selects in-sample actions that adhere to the advantage condition $\xi$ determined by the critic. Therefore, the suboptimal samples with low advantage condition $\xi$ will not disrupt the optimization of optimal policy $\pi_\omega(\cdot|c^*)$. And they enforce valid constraints on the corresponding policy $\pi_\omega(\cdot|c)$, as shown in the lower part of Figure 2. It should be noted that the decoder $p_\psi$ is fixed during both policy evaluation and improvement.

Our A2PO implementation selects TD3+BC [9] as the base backbone for its robustness. The general framework derived above is thoroughly described in Appendix A.

## 5 Experiments

To illustrate the effectiveness of the proposed A2PO method, we conduct experiments on the D4RL benchmark [8]. We aim to answer the following questions: (1) Can A2PO outperform the advanced offline RL methods in both the single-quality datasets and mixed-quality datasets? (Section 5.2) (2) How do different components of A2PO contribute to the overall performance? (Section 5.3 and Appendix D–G) (3) How does A2PO perform under mixed-quality datasets with varying single-quality samples? (Section 5.5 and Appendix H) (4) Can the A2PO agent effectively estimate the quality of different transitions? (Section 5.4 and Appendix I) (5) How does the time overhead of A2PO compare to other baselines? (Section 5.6)

### 5.1 Experiment Settings

**Tasks and Datasets.** We consider four different domains of tasks in D4RL benchmark [8]: Gym, Maze, Adroit, and Kitchen. Each domain contains several tasks and corresponding distinct datasets. We conduct experiments for each Gym task using single-quality and mixed-quality datasets. The single-quality datasets are generated with the *medium* behavior policy. The mixed-quality datasets are the combinations of *random*, *medium*, and *expert* single-quality datasets, including *medium-expert*, *medium-replay*, *random-medium*, *medium-expert*, and *random-medium-expert*. The D4RL benchmark only includes the first two mixed-quality datasets. Thus, following Hong et al. [16, 15], we manually construct the last three mixed-quality datasets by combining the corresponding single-quality datasets in D4RL with equal proportions. For the other domains of tasks, the corresponding D4RL datasets exhibit a significant level of diversity to evaluate the effectiveness of our A2PO algorithm.

**Comparison Methods and Hyperparameters.** We compare the proposed A2PO to several advanced offline RL methods: BCQ [11], TD3+BC [9], CQL [20], EQL [50], especially the advantage-weighted offline RL methods: AWAC [29], IQL [18], CQL+AW [15], LAPO [5]. Besides, we also select the vanilla BC method [33], the model-based offline RL method MOPO [53], and the emerging diffusion-based method Diffusion-QL [47], for comparison. We report the performance of baselines using the best results reported from their own paper. More comparison results can be found in Appendix C. The detailed hyperparameters of A2PO are given in Appendix B.2.

### 5.2 Comparison on D4RL Benchmarks

**Results for Gym Tasks.** The experimental results of all compared methods in the D4RL Gym tasks are presented in Table 1. For the single-quality *medium* dataset and mixed-quality *medium-expert* and *medium-replay* datasets from D4RL, our A2PO achieves state-of-the-art results with low variance. Meanwhile, both conventional offline RL approaches like EQL and advantage-weighted approaches like LAPO still learn acceptable policy, indicating that the conflict issue hardly occurs in these datasets with low diversity. However, the newly constructed mixed-quality datasets, namely *random-medium*, *random-expert*, and *random-medium-expert*, highlight the issue of substantial gaps between behavior policies. The results on these datasets reveal a significant drop in performance for all other baselines.

Table 1: Test returns of our proposed A2PO and baselines on the Gym tasks. $\pm$ corresponds to one standard deviation of the performance on 5 random seeds. The performance is measured by the normalized scores at the last training iteration. **Bold** indicates the best performance in each task.

| Source | Task | BC | BCQ | TD3+BC | CQL | MOPO | EQL | Diffusion-QL | AWAC | IQL | CQL+AW | LAPO | A2PO (Ours) |
|---|---|---|---|---|---|---|---|---|---|---|---|---|---|
| medium | halfcheetah | 42.6 | 47.0 | 48.3 | 44.0 | 42.3 | 47.2 | **51.1** | 43.5 | 47.4 | 49.0 | 46.0 | 47.1±0.2 |
| | hopper | 52.9 | 56.7 | 59.3 | 58.5 | 28.0 | 74.6 | **90.5** | 57.0 | 66.3 | 71.0 | 51.6 | 80.3±4.0 |
| | walker2d | 75.3 | 72.6 | 83.7 | 72.5 | 17.8 | 83.2 | **87.0** | 72.4 | 78.3 | 83.0 | 80.8 | 84.9±0.2 |
| medium replay | halfcheetah | 36.6 | 40.4 | 44.6 | 45.5 | 53.1 | 44.5 | **47.8** | 40.5 | 44.2 | 47.0 | 41.9 | 44.8±0.2 |
| | hopper | 18.1 | 53.3 | 60.9 | 95.0 | 67.5 | 98.1 | 101.3 | 37.2 | 94.7 | 99.0 | 50.1 | **101.6±1.3** |
| | walker2d | 26.0 | 52.1 | 81.8 | 81.6 | 39.0 | 76.6 | **95.5** | 27.0 | 73.9 | 87.0 | 60.6 | 82.8±1.7 |
| medium expert | halfcheetah | 55.2 | 89.1 | 90.7 | 91.6 | 63.3 | 90.6 | **96.8** | 42.8 | 86.7 | 84.0 | 94.2 | 95.6±0.5 |
| | hopper | 52.5 | 81.8 | 98.0 | 105.4 | 23.7 | 105.5 | 111.1 | 55.8 | 91.5 | 91.0 | 111.0 | **113.4±0.5** |
| | walker2d | 107.5 | 109.0 | 110.1 | 108.8 | 44.6 | 110.2 | 110.1 | 74.5 | 109.6 | 109.0 | 110.9 | **112.1±0.2** |
| random medium | halfcheetah | 2.3 | 12.7 | 47.7 | 31.9 | **52.7** | 42.3 | 48.4 | 46.5 | 42.2 | 46.5 | 18.5 | 48.5±0.3 |
| | hopper | 23.2 | 9.2 | 7.4 | 3.3 | 19.9 | 1.7 | 6.9 | 19.5 | 6.2 | 22.6 | 4.2 | **62.1±2.8** |
| | walker2d | 19.2 | 0.2 | 10.7 | 0.2 | 40.2 | 31.4 | 3.3 | 0.0 | 54.6 | 82.0 | 23.6 | **82.3±0.4** |
| random expert | halfcheetah | 13.7 | 2.1 | 43.1 | 15.0 | 18.5 | 47.4 | 86.1 | 87.3 | 28.6 | 80.7 | 52.6 | **90.3±1.6** |
| | hopper | 10.1 | 8.5 | 78.8 | 7.8 | 17.2 | 68.6 | 102.0 | 84.7 | 58.5 | 109.6 | 82.3 | **112.5±1.3** |
| | walker2d | 14.7 | 0.6 | 7.0 | 0.3 | 4.6 | 9.1 | 56.3 | 11.7 | 90.9 | 108.6 | 0.4 | **109.1±1.4** |
| random medium expert | halfcheetah | 2.3 | 15.9 | 62.3 | 13.5 | 26.7 | 42.8 | 81.2 | 2.3 | 61.6 | 76.8 | 71.1 | **90.6±1.6** |
| | hopper | 27.4 | 4.0 | 60.5 | 9.4 | 13.3 | 72.4 | 70.1 | 8.6 | 57.9 | 71.8 | 66.6 | **107.8±0.4** |
| | walker2d | 24.6 | 2.4 | 15.7 | 0.1 | 56.4 | 61.0 | 56.6 | -0.4 | 90.8 | 58.3 | 60.4 | **97.7±6.7** |
| **Gym Total** | | 604.2 | 657.6 | 1010.6 | 784.4 | 628.8 | 1107.2 | 1302.1 | 710.9 | 1183.9 | 1376.9 | 1026.8 | **1563.3** |

Table 2: Test returns of our proposed A2PO and baselines on the Maze, Kitchen, and Adroit tasks.

| Task | BC | BCQ | TD3+BC | CQL | MOPO | EQL | Diffusion-QL | AWAC | IQL | CQL+AW | LAPO | A2PO (Ours) |
|---|---|---|---|---|---|---|---|---|---|---|---|---|
| maze2d-umaze | 0.5 | 24.8 | 24.2 | 5.7 | -15.4 | 56.5 | 66.7 | 94.5 | 56.2 | 19.6 | 78.0 | **133.3±9.6** |
| maze2d-medium | 0.7 | 22.5 | 33.5 | 5.0 | 19.0 | 36.3 | 100.6 | 31.4 | 25.7 | 22.6 | 43.2 | **114.9±12.9** |
| maze2d-large | 1.1 | 43.0 | 128.5 | 12.5 | -0.5 | 57.0 | 116.3 | 43.9 | 45.7 | 10.3 | 69.7 | **156.4±5.8** |
| antmaze-umaze-diverse | 45.6 | 55.0 | 71.4 | **84.0** | 0.0 | 50.8 | 66.2 | 49.3 | 62.2 | 54.0 | 0.0 | 72.6±10.2 |
| antmaze-medium-diverse | 0.0 | 0.0 | 3.0 | 53.7 | 0.0 | 62.2 | 78.6 | 0.7 | 70.0 | 24.0 | 30.2 | **80.2±4.0** |
| antmaze-large-diverse | 0.0 | 2.2 | 0.0 | 14.9 | 0.0 | 38.0 | **56.6** | 1.0 | 47.5 | 40.0 | 22.3 | 52.1±7.9 |
| **Maze Total** | 47.9 | 147.5 | 260.6 | 175.8 | 3.1 | 300.8 | 485.0 | 220.8 | 307.3 | 170.5 | 243.4 | **609.5** |
| kitchen-complete | 33.8 | 8.1 | 0.8 | 43.8 | 40.1 | 70.3 | **84.0** | 3.8 | 62.5 | 30.2 | 53.2 | 69.2±4.9 |
| kitchen-partial | 33.8 | 18.9 | 0.0 | 49.8 | 6.7 | 74.5 | 60.5 | 0.3 | 46.3 | 36.0 | 53.7 | **75.8±2.4** |
| kitchen-mixed | 47.5 | 10.6 | 0.8 | 51.0 | 17.3 | 55.6 | 62.6 | 0.0 | 51.0 | 50.5 | 62.4 | 64.2±3.1 |
| **Kitchen Total** | 115.1 | 37.6 | 1.6 | 144.6 | 64.1 | 200.4 | 207.1 | 4.1 | 159.8 | 116.7 | 169.3 | **209.2** |
| pen-human | 34.4 | 12.3 | -3.7 | 37.5 | 54.6 | 44.3 | **72.8** | 4.3 | 71.5 | -3.0 | 68.1 | 68.9±5.9 |
| pen-cloned | 56.9 | 28.0 | 1.7 | 39.2 | 10.7 | 46.9 | 57.3 | -0.8 | 37.3 | -2.5 | 55.8 | **85.0±7.3** |
| **Adroit Total** | 91.3 | 40.3 | -2.0 | 76.7 | 65.3 | 91.2 | 130.1 | 3.5 | 108.8 | -5.5 | 123.9 | **153.9** |

Instead, our A2PO continues to achieve the best performance on these datasets. When considering the total scores across all datasets, A2PO outperforms the next best-performing AW method, CQL+AW, by over 21%. The results reveal the exceptional ability of A2PO to capture and utilize high-quality interactions within the dataset in order to enforce a reasonable advantage-aware policy constraint and further obtain an optimal agent policy.

**Results for Maze, Kitchen, and Adroit Tasks.** Table 2 presents the experimental results of all the compared methods on the D4RL Maze, Kitchen, and Adroit tasks. The D4RL datasets for these tasks exhibit varying patterns in behavior policy samples. For instance, the Antmaze datasets are highly sub-optimal, while the Adroit datasets have a narrow state-action distribution. Among the offline RL baselines and AW methods, A2PO delivers remarkable performance in these challenging tasks and showcases the robust representation capabilities of the advantage-aware policy.

## 5.3 Ablation Analysis

**Different Advantage condition during training.** The performance comparison of different advantage condition computing methods for agent training is given in Figure 3. Eq. 4 obtains continuous advantage condition $\xi$ in the range of $(-1, 1)$. To evaluate the effectiveness of the continuous computing method, we design a discrete form of advantage condition: $\xi_{\text{dis}} = \text{sgn}(\xi) \cdot \mathbf{1}_{|\xi|>\epsilon}$, where $\text{sgn}(\cdot)$ is the symbolic function, and $\mathbf{1}_{|\xi|>\epsilon}$ is the indicator function returning 1 if the absolute value of $\xi$ is greater than the hyperparameter of threshold $\epsilon$, otherwise 0. Thus, the advantage condition $\xi_{\text{dis}}$ is constrained to discrete value of $\{-1, 0, 1\}$. Another special form of advantage condition is $\xi_{\text{fix}} = 1$ for all state-action pairs, in which the advantage-aware ability is lost. Figure 3a shows that setting

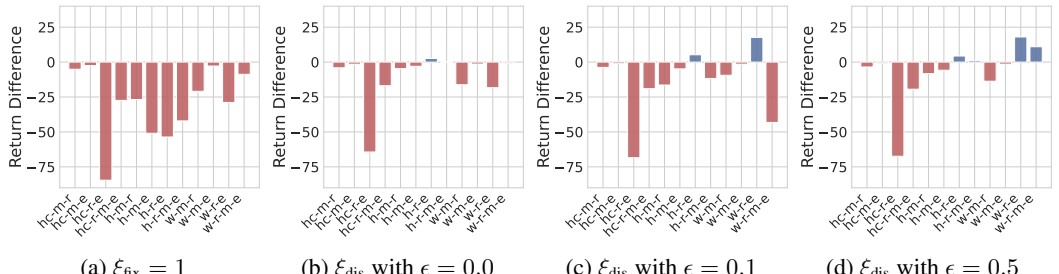

(a) $\xi_{\text{fix}} = 1$  (b) $\xi_{\text{dis}}$ with $\epsilon = 0.0$  (c) $\xi_{\text{dis}}$ with $\epsilon = 0.1$  (d) $\xi_{\text{dis}}$ with $\epsilon = 0.5$

Figure 3: Test return difference of A2PO with different discrete advantage conditions during training compared with original A2PO with continuous advantage condition during training. Task abbreviations are listed in Appendix B.1. Test returns are reported in Appendix D.

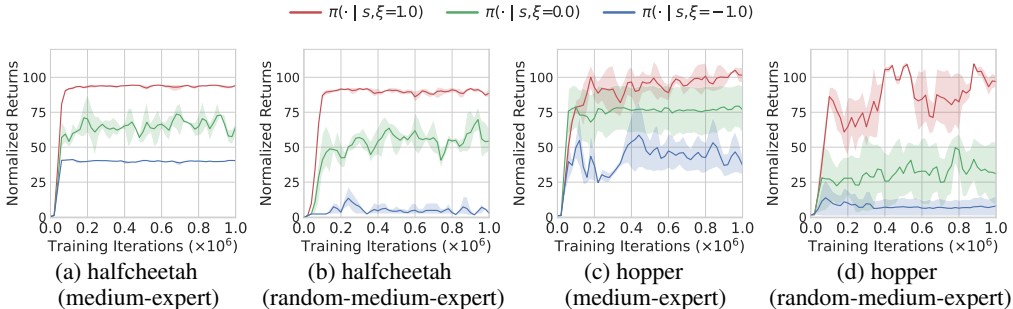

(a) halfcheetah (medium-expert)  (b) halfcheetah (random-medium-expert)  (c) hopper (medium-expert)  (d) hopper (random-medium-expert)

Figure 4: Learning curves of A2PO under different fixed advantage inputs during the test while using the original continuous advantage condition for training. Test returns are reported in Appendix E.

$\xi_{\text{fix}} = 1$ without explicitly advantage-aware mechanism leads to a significant performance decreasing, especially in the new mixed-quality dataset. Meanwhile, $\xi_{\text{dis}}$ with different values of threshold $\epsilon$ achieve slightly inferior results than the continuous $\xi$. This outcome strongly supports the efficiency of continuous $\xi$. Although the $\xi_{\text{dis}}$ signals are more stable, $\xi_{\text{dis}}$ hidden the concrete advantage value, causing a mismatch between the advantage value and the sampled transition.

**Different Advantage Condition for Test.** The performance comparison of different discrete advantage conditions input for the test is given in Figure 4. To ensure clear differentiation, we select $\xi$ from $\{-1, 0, 1\}$. The different designated advantage conditions $\xi$ are fixed input for the actor, leading to different policies $\pi_\omega(\cdot|s,\xi)$. The final outcomes demonstrate the partition of returns corresponding to the policies with different $\xi$. Furthermore, the magnitude of the gap increases as the offline dataset includes samples from more diverse behavior policies. These observations provide strong evidence for the success of A2PO disentangling the behavior policies under the multi-quality dataset.

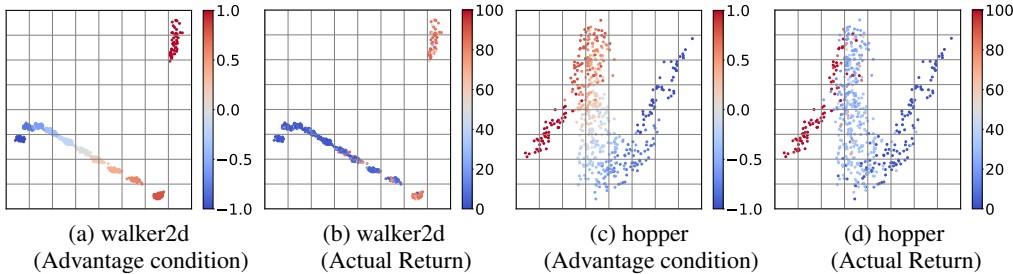

(a) walker2d (Advantage condition)  (b) walker2d (Actual Return)  (c) hopper (Advantage condition)  (d) hopper (Actual Return)

Figure 5: Visualization of A2PO latent representation after applying PCA with different advantage conditions and actual returns in the *walker2d-medium-replay* and *hopper-medium-replay* tasks. Each data point indicates a latent representation $\tilde{z}$ based on the initial state and different advantage conditions sampled uniformly from $[-1, 1]$. The actual return is measured under the corresponding sampled advantage condition. The value magnitude is indicated with varying shades of color.

## 5.4 Visualization

Figure 5 presents the visualization of A2PO latent representation. The uniformly sampled advantage condition $\xi$ combined with the initial state $s$, are fed into the actor-network to get the latent representation generated by the final layer of the actor. The result demonstrates that the representations converge according to the advantage and the actual return. Moreover, upon comparing Figure 5(a,b), as well as Figure 5(c,d) separately, we observe that the latent action representation follows the same alteration pattern based on the actual real return. These observations demonstrate that our advantage-aware policy effectively capture policies with different returns by the designated advantage input $\xi$. This provides compelling evidence for the efficacy of the A2PO policy construction. More experiments of advantage estimation conducted on different tasks and datasets are presented in Appendix I.

## 5.5 Robustness

Figure 6 presents the experimental results of A2PO across mixed-quality datasets with varying proportions of single-quality samples. Following the methodology of [15, 16], we evaluate the effectiveness of A2PO on three mixed-quality datasets: *medium-expert*, *random-medium*, and *random-expert*. These datasets consist of a total number of $1 \times 10^6$ transitions. We vary the proportions $\sigma$ of higher quality samples and $(1-\sigma)$ of lower quality samples. The results demonstrate that our A2PO effectively captures and infers high-quality potential behavior policies for proper policy regularization, even with a small proportion of high-quality samples. Additionally, as the $\sigma$ becomes larger, the variance decreases. Thus, A2PO demonstrates its robustness in handling variations in the proportions of different single-quality samples, guaranteeing consistently high performance.

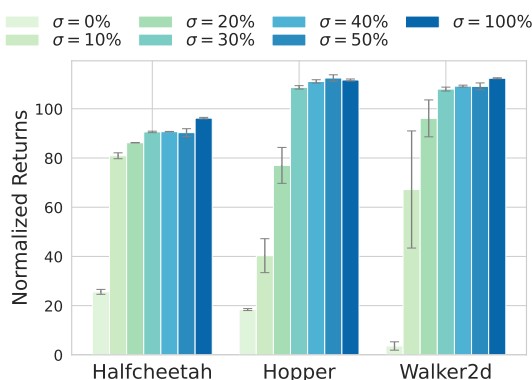

Figure 6: Compare the returns of A2PO under *random-expert* dataset with different high-quality data proportions $\sigma$ in the Gym tasks. Detail returns are reported in Appendix H.

## 5.6 Time Overhead

We measure the training times of A2PO as well as other baselines, which are presented in Table 7. The experiments are performed on a cluster of 4 A40 GPUs under *halfcheetah-medium-expert-v2* scenarios for $1 \times 10^6$ steps. Although the A2PO runtime is longer than the lightweight algorithms like IQL due to CVAE training, our A2PO is more efficient compared to other AW methods such as LAPO and CQL+AW.

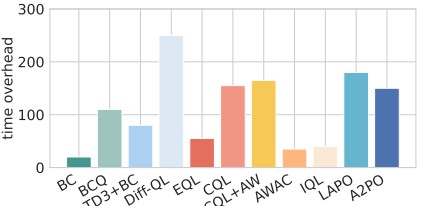

Figure 7: Compare the time overhead of A2PO and other baselines.

## 6 Conclusion

In this paper, we propose a novel approach, termed as A2PO, to tackle the constraint conflict issue on mixed-quality offline datasets with advantage-aware policy constraints. Specifically, A2PO utilizes a CVAE to effectively disentangle the action distributions associated with various behavior policies. This is achieved by modeling the advantage values of all training data as conditional variables. Consequently, advantage-aware agent policy optimization can be focused on maximizing high advantage values while conforming to the disentangled distribution constraint imposed by the mixed-quality dataset. Experimental results show that A2PO successfully decouples the underlying behavior policies and significantly outperforms advanced offline RL competitors. For our future work, we will extend A2PO to multi-task offline RL scenarios characterized by a greater diversity of behavior policies and a more prominent constraint conflict issue.

**Limitations.** The limitation of A2PO is that it incorporates CVAE during training, which may lead to quite a large time overhead. However, the results presented in Section 5.6 show that the time overhead of A2PO remains reasonably acceptable when compared to other baseline methods.

# 7 Acknowledgement

This work was supported in part by the Hangzhou Joint Funds of the Zhejiang Provincial Natural Science Foundation of China under Grant No. LHZSD24F020001, in part by the Fundamental Research Funds for the Central Universities under Grant No. 226-2024-00058, and in part by the Zhejiang Province High-Level Talents Special Support Program "Leading Talent of Technological Innovation of Ten-Thousands Talents Program" under Grant No. 2022R52046.

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

# A Pseudocode

To make the proposed A2PO method clearer for readers, the pseudocode is provided in Algorithm 1.

---

**Algorithm 1** Advantage-Aware Policy Optimization (A2PO)

---

**Input:** offline dataset $\mathcal{D}$, CVAE training step $K$, total training step $T$, soft update rate $\tau$.
**Initialize:** CVAE encoder $q_\varphi$ and decoder $p_\psi$, actor network $\pi_\omega$, critic networks $Q_\theta$ and $V_\phi$.

   **for** $i = 1$ **to** $T$ **do**

      Sample random minibatch of transitions $\mathcal{B} = \{(s, a, r, s')\} \sim \mathcal{D}$.
      Calculate $\xi = \tanh(\min_{i=1,2} Q_{\theta_i}(s, a) - V_\phi(s)), \xi^* = 1, c = s||\xi, c^* = s||\xi^*$.
      # Behavior Policy Disentangling

      **if** $i \leq K$ **then**

         Optimize CVAE encoder $q_\varphi$ and decoder $p_\psi$ according to Eq. 5 as

$$\mathcal{L}_{\text{CVAE}}(\varphi, \psi) = -\mathbb{E}_\mathcal{D} \left[ \mathbb{E}_{q_\varphi(z|a,c)} \left[\log(p_\psi(a|z,c))\right] + \alpha \cdot \text{KL} \left[(q_\varphi(z|a,c) \parallel p(z))\right] \right].$$

      **end if**

      # Agent Policy Optimization

      Optimize critic networks $Q_\theta$ and $V_\phi$ according to Eq. 1 and Eq. 7 as

$$\mathcal{L}_{\text{TD}}(\theta, \phi) = \mathbb{E}_{\substack{(s,a,r,s')\sim\mathcal{D}, \\ \tilde{z}^*\sim\pi_\omega(\cdot|c^*), \\ a_\xi^*\sim p_\psi(\cdot|\tilde{z}^*,c^*)}} \left[ \sum_i \left[r+\gamma \min_j Q_{\hat{\theta}_j}(s', a_\xi^*)-Q_{\theta_i}(s,a)\right]^2 + \left[r+\gamma \min_i Q_{\hat{\theta}_i}(s', a_\xi^*)-V_\phi(s)\right]^2 \right].$$

      Optimize actor network $\pi_\omega$ according to Eq. 8 as

$$\mathcal{L}_{\text{AC}}(\omega) = -\lambda \cdot \mathbb{E}_{\substack{s\sim\mathcal{D}, \\ \tilde{z}^*\sim\pi_\omega(\cdot|c^*), \\ a_\xi^*\sim p_\psi(\cdot|\tilde{z}^*,c^*)}} \left[Q_{\theta_1}(s, a_\xi^*)\right] + \mathbb{E}_{\substack{(s,a)\sim\mathcal{D}, \\ \tilde{z}\sim\pi_\omega(\cdot|c), \\ a_\xi\sim p_\psi(\cdot|\tilde{z},c)}} \left[(a - a_\xi)^2\right].$$

      Update the target networks with $\hat{\theta} \leftarrow (1 - \tau)\hat{\theta} + \tau\theta, \hat{\phi} \leftarrow (1 - \tau)\hat{\phi} + \tau\phi$.

   **end for**

---

# B Experiment Details

## B.1 Task Abbreviations and Task Versions

In order to improve the readability and conciseness, we adopt abbreviations for the Gym tasks throughout the main text. The corresponding abbreviations for each task are provided in Table 3. For the specific task versions, we use the '-v2' version for the Gym tasks, the '-v1' version for the Maze2d tasks, the '-v0' version for the Antmaze tasks, the '-v0' version for the Kitchen tasks, the '-v1' version for the Adroit tasks.

Table 3: The abbreviation of the corresponding Gym task and dataset.

| Dataset | halfcheetah | hopper | walker2d |
|---|---|---|---|
| random | hc-r | h-r | w-r |
| medium | hc-m | h-m | w-m |
| expert | hc-e | h-e | w-e |
| medium-replay | hc-m-r | h-m-r | w-m-r |
| medium-expert | hc-m-e | h-m-e | w-m-e |
| random-medium | hc-r-m | h-r-m | w-r-m |
| random-expert | hc-m-e | h-m-e | w-m-e |
| random-medium-expert | hc-r-m-e | h-r-m-e | w-r-m-e |

**B.2  Implementation Details**

In this section, we provide the implementation details of our experiments. We conducted our experiments using PyTorch 3.8 [30] on a cluster of 4 A40 GPUs.The source code will be made publicly available upon the publication of this paper.

The critic, actor, and CVAE networks are all constructed using a 2-layer MLP with 256 hidden units. They are updated using the Adam optimizer with a learning rate of $3 \times 10^{-4}$. Additionally, we employ a target critic network with a soft-update rate of $5 \times 10^{-3}$. Following the TD+3BC approach [9], we incorporate Q normalization, policy noise, and policy clipping during the training process. or the hyperparameter value $\alpha$ in Q normalization, we adopted the strategy proposed by Wang et al. [47], selecting values of $5.0$ and $6.0$ for the antmaze-medium/large-diverse tasks respectively to promote robust and stable Q Learning. In contrast, we chose $\alpha = 0.005$ for the kitchen-partial task to facilitate policy regularization. For other tasks, setting $\alpha = 1.0$ yields state-of-the-art results. As for the CVAE step $K$, 0.2M steps demonstrate sufficient robustness for most tasks, while the optimal settings for the maze2d-umaze/medium tasks are 0.4M and 0.1M steps, respectively. Among the tasks, the continuous form of the advantage signal $\xi$, as described in Eq. 4, consistently produces outstanding results across most scenarios. However, the advantage distribution may vary significantly across all the tasks, necessitating the fine-tuning of advantage computation methods based on online evaluations. The maze2d-umaze task requires a discrete form with $\epsilon = 0.1$. In the case of the maze2d-medium task, we utilize an $\epsilon$ value of $0.2$. For both the maze2d-large and antmaze-umaze tasks, we increase this to $\epsilon = 0.8$. In contrast, for the kitchen-partial/mixed tasks, we adopt a value of $\epsilon = 0.0$. Further details on this sensitivity can be found in Section 5.3 and Appendix D.

Since all of the original papers of other baselines do not provide the complete results of all the tasks, we make implementation of the baselines. The CQL, IQL and MOPO baselines are implemented using the implementations provided at github.com/young-geng/cql, gwthomas/iql-pytorch, and github.com/yihaosun1124/OfflineRL-Kit, respectively. The remaining baselines, including BCQ, CQL, TD3+BC, EQL, Diffusion-QL, AWAC, CQL+AW, and LAPO, are implemented using the original implementations provided by the authors of the respective papers. These implementations can be found at: BCQ github.com/sfujim/BCQ, TD3+BC github.com/sfujim/TD3_BC, EQL github.com/ryanxhr/IVR, Diffusion-QL github.com/Zhendong-Wang/Diffusion-Policies-for-Offline-RL, AWAC github.com/rail-berkeley/rlkit/tree/master/examples/awac, CQL+AW github.com/Improbable-AI/harness-offline-rl, and LAPO github.com/pcchenxi/LAPO-offlienRL.

## C  Additional Comparisons with More Baselines

In this section, we have added the data rebalance baseline ReD [54]; novel policy-constrain baseline PRDC [37] and return-conditioned baselines Decision Transformer and %BC [3] as the comparison baselines to further evaluate the superiority of our A2PO. The results are illustrated in Table 4. Our A2PO method demonstrates comparable performance to various offline RL methods. While the performance of A2PO is slightly inferior to that of PRDC in the *medium* and *medium-replay* dataset, it surpasses all additional baselines in the majority of scenarios.

## D  Analysis of Advantage Condition Input for Training

In this section, we provide a full comparison of different advantage condition computing methods for training on Gym tasks in Table 5. These computing methods are thoroughly described in Section 5.3. The A2PO algorithm degenerates into TD3+BC under $\xi_{\text{fix}} = 1$, which treats each sample constraint equally with no regard for the differences in the data quality. In this case, it achieves significantly worse performance compared to using varied $\xi$ values, particularly on the *random-expert* datasets which highlight the substantial gap between the behavior policies. This observation highlights the potential risks of constraint conflict issues in policy regularization methods. On the other hand, the discrete $\xi_{\text{dis}}$ yields slightly inferior performances and larger variances compared to the continuous $\xi$, when different threshold values $\epsilon$ are used. While the $\xi_{\text{dis}}$ signals demonstrate more stability, the discretization process fails to accurately capture the advantage value associated with each sampled transition. Consequently, there is a mismatch between the advantage value and the actual sampled transition, which negatively impacts the overall algorithm performance. However, by utilizing the

Table 4: Test returns of our proposed A2PO and additional baselines on the locomotion tasks. The *italic* results of the baselines are obtained from the corresponding original paper.

| Source | Task | %BC | DT | IQL+ReD | PRDC | A2PO (Ours) |
|---|---|---|---|---|---|---|
| medium | halfcheetah | *42.5* | *42.6* | *47.6* | ***63.5*** | $47.1_{\pm0.2}$ |
| | hopper | *56.9* | *67.6* | *66.0* | ***100.3*** | $80.3_{\pm4.0}$ |
| | walker2d | *75.0* | *74.0* | *78.6* | ***85.2*** | $84.9_{\pm0.2}$ |
| medium replay | halfcheetah | *40.6* | *36.6* | *44.3* | ***55.0*** | $44.8_{\pm0.2}$ |
| | hopper | *75.9* | *82.7* | *101.0* | *100.1* | $\mathbf{101.6_{\pm1.3}}$ |
| | walker2d | *62.5* | *66.6* | *79.5* | ***92.0*** | $82.8_{\pm1.7}$ |
| medium expert | halfcheetah | *92.9* | *86.8* | *92.6* | *94.5* | $\mathbf{95.6_{\pm0.5}}$ |
| | hopper | *110.9* | *82.7* | *101.0* | *109.2* | $\mathbf{113.4_{\pm0.5}}$ |
| | walker2d | *109.0* | *108.1* | *110.5* | *111.2* | $\mathbf{112.1_{\pm0.2}}$ |
| random medium | halfcheetah | 40.1 | 42.0 | 42.0 | **56.5** | $48.5_{\pm0.3}$ |
| | hopper | 21.0 | 3.1 | 6.7 | 5.5 | $\mathbf{62.1_{\pm2.8}}$ |
| | walker2d | 32.0 | 66.9 | 60.0 | 5.5 | $\mathbf{82.3_{\pm0.4}}$ |
| random expert | halfcheetah | 7.7 | 10.3 | 42.4 | 1.3 | $\mathbf{90.3_{\pm1.6}}$ |
| | hopper | 2.4 | 90.2 | 16.7 | 24.8 | $\mathbf{112.5_{\pm1.3}}$ |
| | walker2d | 53.4 | 103.4 | 93.2 | 1.1 | $\mathbf{109.1_{\pm1.4}}$ |
| random medium expert | halfcheetah | 29.1 | 42.6 | 39.1 | 10.5 | $\mathbf{90.6_{\pm1.6}}$ |
| | hopper | 62.0 | 46.1 | 31.3 | 88.5 | $\mathbf{107.8_{\pm0.4}}$ |
| | walker2d | 10.6 | 78.8 | 52.0 | 4.9 | $\mathbf{97.7_{\pm6.7}}$ |

Table 5: Test returns of our proposed A2PO with different advantage conditions during training. $\pm$ corresponds to one standard deviation of the average evaluation of the performance on 5 random seeds. The performance is measured by the normalized scores at the last training iteration. **Bold** indicates the best performance in each task.

| Source | Task | $\xi_{\text{fix}} = 1$ | $\xi_{\text{dis}}, \epsilon = 0.0$ | $\xi_{\text{dis}}, \epsilon = 0.1$ | $\xi_{\text{dis}}, \epsilon = 0.5$ | Continuous $\xi$ |
|---|---|---|---|---|---|---|
| medium replay | halfcheetah | $39.6_{\pm0.6}$ | $40.7_{\pm0.8}$ | $41.0_{\pm0.5}$ | $41.2_{\pm0.5}$ | $\mathbf{44.8_{\pm0.2}}$ |
| | hopper | $74.8_{\pm11.2}$ | $96.9_{\pm1.7}$ | $85.2_{\pm6.0}$ | $93.3_{\pm4.4}$ | $\mathbf{101.6_{\pm1.3}}$ |
| | walker2d | $61.9_{\pm1.6}$ | $63.6_{\pm5.6}$ | $73.4_{\pm5.8}$ | $69.1_{\pm6.7}$ | $\mathbf{82.8_{\pm1.7}}$ |
| medium expert | halfcheetah | $93.1_{\pm1.5}$ | $94.1_{\pm0.4}$ | $94.5_{\pm0.5}$ | $94.8_{\pm0.0}$ | $\mathbf{95.6_{\pm0.5}}$ |
| | hopper | $62.5_{\pm5.5}$ | $110.5_{\pm0.7}$ | $108.6_{\pm1.5}$ | $107.6_{\pm2.9}$ | $\mathbf{113.4_{\pm0.5}}$ |
| | walker2d | $109.3_{\pm0.1}$ | $110.7_{\pm0.3}$ | $110.7_{\pm0.1}$ | $110.7_{\pm6.7}$ | $\mathbf{112.1_{\pm0.2}}$ |
| random expert | halfcheetah | $5.8_{\pm1.1}$ | $26.0_{\pm6.4}$ | $21.9_{\pm6.2}$ | $22.9_{\pm3.7}$ | $\mathbf{90.3_{\pm1.6}}$ |
| | hopper | $51.6_{\pm31.8}$ | $107.8_{\pm3.7}$ | $110.4_{\pm1.0}$ | $95.8_{\pm19.4}$ | $\mathbf{112.5_{\pm1.3}}$ |
| | walker2d | $63.1_{\pm33.1}$ | $73.6_{\pm52.0}$ | $\mathbf{109.6_{\pm0.0}}$ | $109.9_{\pm0.1}$ | $109.1_{\pm1.4}$ |
| random medium expert | halfcheetah | $63.2_{\pm1.2}$ | $73.7_{\pm5.4}$ | $71.6_{\pm2.7}$ | $71.2_{\pm3.7}$ | $\mathbf{90.6_{\pm1.6}}$ |
| | hopper | $65.7_{\pm38.2}$ | $108.0_{\pm1.2}$ | $96.1_{\pm15.8}$ | $\mathbf{108.0_{\pm0.1}}$ | $107.8_{\pm0.4}$ |
| | walker2d | $89.0_{\pm13.5}$ | $96.9_{\pm13.8}$ | $54.5_{\pm54.6}$ | $\mathbf{108.7_{\pm3.3}}$ | $97.7_{\pm6.7}$ |

continuous advantage value, our A2PO is able to better adapt to the varying environment dynamics and complexities found in the mixed-quality dataset, thereby further getting state-of-the-art performance with low variance. This enables a more precise capture of the relationship between the advantage condition and the disentangled behavior policies.

# E  Analysis of Advantage Condition Input for Testing

In this section, we provide a full comparison of different fixed advantage condition inputs for testing on Gym tasks in Table 6 as a supplement for Figure 4. It should be noted that in contrast to the ablation study shown in Appendix D which investigates different methods for computing advantages

Table 6: Test returns of A2PO with different discrete advantage conditions for test while using the original continuous advantage condition during training.

| Source | Task | $\pi_\omega(\cdot\|s,\xi\!=\!-1)$ | $\pi_\omega(\cdot\|s,\xi\!=\!0)$ | $\pi_\omega(\cdot\|s,\xi\!=\!1)$ |
|---|---|---|---|---|
| expert | halfcheetah | $87.2_{\pm0.3}$ | $93.2_{\pm0.4}$ | $\mathbf{96.3}_{\pm0.3}$ |
| | hopper | $84.9_{\pm22.2}$ | $106.6_{\pm6.7}$ | $\mathbf{111.7}_{\pm10.4}$ |
| | walker2d | $7.9_{\pm3.1}$ | $48.9_{\pm12.7}$ | $\mathbf{112.4}_{\pm0.2}$ |
| medium replay | halfcheetah | $4.7_{\pm5.6}$ | $21.5_{\pm8.6}$ | $\mathbf{44.8}_{\pm0.2}$ |
| | hopper | $11.4_{\pm6.3}$ | $25.3_{\pm1.1}$ | $\mathbf{101.6}_{\pm1.3}$ |
| | walker2d | $2.0_{\pm3.4}$ | $22.5_{\pm3.7}$ | $\mathbf{82.8}_{\pm1.7}$ |
| medium expert | halfcheetah | $40.4_{\pm0.6}$ | $64.5_{\pm4.6}$ | $\mathbf{95.6}_{\pm0.5}$ |
| | hopper | $37.1_{\pm16.2}$ | $76.9_{\pm24.4}$ | $\mathbf{113.4}_{\pm0.5}$ |
| | walker2d | $5.8_{\pm0.9}$ | $73.0_{\pm5.8}$ | $\mathbf{112.1}_{\pm0.2}$ |
| random expert | halfcheetah | $2.8_{\pm4.2}$ | $79.9_{\pm11.9}$ | $\mathbf{90.3}_{\pm1.6}$ |
| | hopper | $0.8_{\pm0.0}$ | $11.5_{\pm9.1}$ | $\mathbf{112.5}_{\pm1.3}$ |
| | walker2d | $-0.1_{\pm0.0}$ | $3.14_{\pm4.5}$ | $\mathbf{109.1}_{\pm1.4}$ |
| random medium expert | halfcheetah | $2.9_{\pm1.9}$ | $54.3_{\pm7.0}$ | $\mathbf{90.6}_{\pm1.6}$ |
| | hopper | $7.9_{\pm7.2}$ | $31.0_{\pm19.1}$ | $\mathbf{107.8}_{\pm0.4}$ |
| | walker2d | $1.5_{\pm1.6}$ | $9.2_{\pm5.9}$ | $\mathbf{97.7}_{\pm6.7}$ |

during CVAE and agent learning, this section utilizes the original continuous advantage computation method. However, the designated $\xi$ is varied during the testing process. On one hand, the CVAE decoder $p_\psi$ can generate CVAE-approximated behavior policies by varying the input $\xi$. On the other hand, Eq. 8 for policy improvement regulates the advantage-aware policy to closely resemble samples from the same advantage condition. This indicates that the quality of the generated action should be positively correlated with the specified action input. The experimental outcomes demonstrate the relationship between varying values of $\xi$ and the corresponding A2PO policy returns: the agent performance consistently improves as the fixed advantage condition input $\xi$ increases. Moreover, the gap between the returns increases as the offline dataset includes more diverse behavior policies. These observations serve as strong evidence for the effectiveness of A2PO in effectively disentangling the behavior policies within a multi-quality dataset.

## F  Analysis of the CVAE Policy

The CVAE policy corresponds to the CVAE decoder $p_\psi(a|z_0, c^*)$, where $z_0$ is sampled from $\mathcal{N}(0,1)$, and $c^* = s\,\|\,\xi^*$. As described in Section 4.1, the advantage-aware CVAE utilizes the advantage condition computed by the agent critic to construct the ELBO loss, which is formulated as advantage-guided supervised learning [12]. The CVAE decoder $p_\psi(a|z_0, c^*)$ generates action outputs of varying qualities based on the target advantage input signal $\xi$. By conditioning on the maximum normalized advantage value $\xi^*$ to get the optimal action of CVAE and agent policy, we present thorough comparison results of the CVAE policy and agent policy in Table 7. The performance of the CVAE policy indicates that it only demonstrates superior performance in a limited set of tasks and datasets, specifically the *hopper-random* and *walker2d-random* environments. The A2PO agent consistently outperforms the CVAE agent in the majority of cases. This performance gap is particularly significant in the newly constructed mix-quality datasets including *random-expert* and *random-medium-expert* highlighting the conflict issue. These findings suggest that A2PO achieves well-disentangled behavior policies and an optimal agent policy, surpassing the capabilities of CVAE-reconstructed behavior policies.

## G  Analysis of A2PO Policy Optimization

In this section, we evaluate the effectiveness of the policy regularization term within the policy improvement step as shown in Equation 8. Previous approaches [56, 5] that construct agent policies within the CVAE latent space do not incorporate the BC regularization term, as the latent action

Table 7: Test returns of A2PO with CVAE policy or agent policy.

| Source | Task | CVAE policy $p_\psi(\cdot|z_0, c^*)$ | Agent Policy $\pi(\cdot|\tilde{z}^*, c^*)$ |
|---|---|---|---|
| medium | halfcheetah | $45.7_{\pm0.3}$ | $\mathbf{47.1}_{\pm0.2}$ |
|  | hopper | $57.1_{\pm2.8}$ | $\mathbf{80.3}_{\pm4.0}$ |
|  | walker2d | $81.9_{\pm0.7}$ | $\mathbf{84.9}_{\pm0.2}$ |
| medium replay | halfcheetah | $39.2_{\pm1.8}$ | $\mathbf{44.8}_{\pm0.2}$ |
|  | hopper | $91.5_{\pm11.4}$ | $\mathbf{101.6}_{\pm1.3}$ |
|  | walker2d | $63.4_{\pm9.5}$ | $\mathbf{82.8}_{\pm1.7}$ |
| medium expert | halfcheetah | $93.4_{\pm0.9}$ | $\mathbf{95.6}_{\pm0.5}$ |
|  | hopper | $112.2_{\pm0.6}$ | $\mathbf{113.4}_{\pm0.5}$ |
|  | walker2d | $110.5_{\pm0.3}$ | $\mathbf{112.1}_{\pm0.2}$ |
| random medium | halfcheetah | $41.1_{\pm0.9}$ | $\mathbf{48.5}_{\pm0.3}$ |
|  | hopper | $15.5_{\pm11.7}$ | $\mathbf{62.1}_{\pm2.8}$ |
|  | walker2d | $41.9_{\pm6.0}$ | $\mathbf{82.3}_{\pm0.4}$ |
| random expert | halfcheetah | $36.9_{\pm15.9}$ | $\mathbf{90.3}_{\pm1.6}$ |
|  | hopper | $81.4_{\pm15.6}$ | $\mathbf{112.5}_{\pm1.3}$ |
|  | walker2d | $-0.1_{\pm0.1}$ | $\mathbf{109.1}_{\pm1.4}$ |
| random medium expert | halfcheetah | $66.2_{\pm6.0}$ | $\mathbf{90.6}_{\pm1.4}$ |
|  | hopper | $56.7_{\pm7.8}$ | $\mathbf{107.8}_{\pm0.4}$ |
|  | walker2d | $22.7_{\pm6.1}$ | $\mathbf{97.7}_{\pm6.7}$ |

space inherently imposes constraints on the action outputs. In contrast, our advantage-aware policy integrates the BC term into the actor loss to explicitly ensure appropriate pessimism. Meanwhile, to guarantee the policy action outputs are consistent with the advantage condition $\xi$, we develop a variant of A2PO, designated A2PO$[a_{\mathrm{cvae}}^*]$, which constrains the agent output to the optimal CVAE policy $p_\psi(\cdot|z_0 \sim \mathcal{N}(0, I), c^*)$. Unlike this variant, the original A2PO utilizes the MSE loss to align the agent action output $\pi(\cdot|\tilde{z}^*, c^*)$ with the sampled action $a$. To assess the impact of our A2PO architecture and the BC regularization, we conduct an ablation study by comparing the results of different A2PO variants and LAPO, as demonstrated in Table 8. The results indicate that even without the BC regularization term, A2PO consistently outperforms LAPO in most tasks. Furthermore, the original formulation of the BC term in the main text can enhance performance in most cases. This comparison highlights the superior performance achieved by A2PO, showcasing its effective disentanglement of action distributions from different behavior policies to enforce a reasonable advantage-aware policy constraint and obtain an optimal agent policy.

## H  Analysis of A2PO Under Mixed-quality Dataset with Different Proportion

In this section, we conducted additional experiments to assess the robustness of A2PO under various mixed-quality datasets. The mixed datasets consist of a fixed number of $1 \times 10^6$ offline samples. These datasets are generated by combining $\sigma$ of either an expert or medium dataset (high-return) and $1 - \sigma$ of a random or medium dataset (low-return). The quantitative results are presented in Table 9. The results demonstrate that our A2PO algorithm is capable of achieving expert-level performance even with a limited number of high-quality samples. Moreover, as the proportion $\sigma$ increases, the A2PO policy becomes more stable. These results indicate the robustness and effectiveness of our A2PO algorithm in deriving optimal policies across diverse structures of offline datasets.

## I  Advantage Visualization

In this section, we illustrate the distribution of initial state-action pairs in offline datasets across various tasks, along with the corresponding actual return and estimated advantage values. Figure 8 and 9 present a comparative analysis of the actual return, LAPO, and our A2PO advantage approximation. The findings indicate that LAPO exhibits limited discrimination in assessing transition advantages,

Table 8: Test returns of LAPO, A2PO and its variants in relation to policy regularization.

| Source | Task | LAPO | A2PO w/o BC | A2PO$[a^*_{\text{cvae}}]$ | A2PO (Ours) |
|---|---|---|---|---|---|
| medium | halfcheetah | $46.0_{\pm0.1}$ | $46.8_{\pm0.1}$ | $46.6_{\pm0.1}$ | $\mathbf{47.1}_{\pm0.2}$ |
| | hopper | $51.6_{\pm2.6}$ | $70.1_{\pm4.0}$ | $71.9_{\pm9.4}$ | $\mathbf{80.3}_{\pm4.0}$ |
| | walker2d | $80.8_{\pm1.3}$ | $82.0_{\pm1.1}$ | $82.0_{\pm0.7}$ | $\mathbf{84.9}_{\pm0.2}$ |
| medium replay | halfcheetah | $41.9_{\pm0.5}$ | $42.0_{\pm0.3}$ | $40.3_{\pm1.2}$ | $\mathbf{44.8}_{\pm0.2}$ |
| | hopper | $50.1_{\pm11.2}$ | $96.5_{\pm1.5}$ | $96.1_{\pm2.0}$ | $\mathbf{101.6}_{\pm1.3}$ |
| | walker2d | $60.6_{\pm10.5}$ | $71.1_{\pm8.0}$ | $79.9_{\pm1.6}$ | $\mathbf{82.8}_{\pm1.7}$ |
| medium expert | halfcheetah | $94.2_{\pm0.5}$ | $94.3_{\pm0.0}$ | $94.8_{\pm0.3}$ | $\mathbf{95.6}_{\pm0.5}$ |
| | hopper | $111.0_{\pm0.4}$ | $107.3_{\pm2.0}$ | $87.7_{\pm19.0}$ | $\mathbf{113.4}_{\pm0.5}$ |
| | walker2d | $110.9_{\pm0.2}$ | $111.6_{\pm0.1}$ | $111.5_{\pm0.3}$ | $\mathbf{112.1}_{\pm0.2}$ |
| random expert | halfcheetah | $52.6_{\pm17.3}$ | $31.4_{\pm6.3}$ | $\mathbf{93.1}_{\pm6.2}$ | $90.3_{\pm1.6}$ |
| | hopper | $82.3_{\pm19.0}$ | $\mathbf{113.2}_{\pm1.2}$ | $81.8_{\pm0.2}$ | $112.5_{\pm1.3}$ |
| | walker2d | $0.4_{\pm0.5}$ | $66.8_{\pm11.0}$ | $96.8_{\pm3.3}$ | $\mathbf{109.1}_{\pm1.4}$ |
| random medium | halfcheetah | $18.5_{\pm1.0}$ | $43.2_{\pm0.5}$ | $41.0_{\pm1.6}$ | $\mathbf{48.5}_{\pm0.3}$ |
| | hopper | $4.2_{\pm3.1}$ | $25.7_{\pm9.2}$ | $40.9_{\pm2.0}$ | $\mathbf{62.1}_{\pm2.8}$ |
| | walker2d | $23.6_{\pm34.0}$ | $72.3_{\pm4.4}$ | $57.8_{\pm3.4}$ | $\mathbf{82.3}_{\pm0.4}$ |
| random medium expert | halfcheetah | $71.1_{\pm0.4}$ | $70.8_{\pm4.2}$ | $89.0_{\pm4.8}$ | $\mathbf{90.6}_{\pm1.6}$ |
| | hopper | $66.6_{\pm19.3}$ | $86.5_{\pm7.3}$ | $12.8_{\pm4.0}$ | $\mathbf{107.8}_{\pm0.4}$ |
| | walker2d | $60.4_{\pm43.2}$ | $\mathbf{110.4}_{\pm1.2}$ | $63.0_{\pm4.5}$ | $97.7_{\pm6.7}$ |
| Total | | 1026.8 | **1342.0** | 1287.0 | 1563.3 |

while A2PO effectively distinguishes between transitions of varying data quality. These results underscore the limitations of the AW method and highlight the superiority of our A2PO approach.

## J  Boarder Impact

This paper presents an offline advantage-aware learning approach that leverages the estimated advantage condition to deal with mixed-quality datasets. The advantage-aware concept brings a new perspective to the solution of real-world RL tasks, facilitating a more practical and effective utilization of the offline datasets. It has the potential to enhance the robustness of the agent towards the varying offline datasets from real-world RL scenarios, where the pre-collected offline datasets are noisy and often not as well-organized as the D4RL standardized datasets.

Table 9: Test returns of A2PO under *random-expert* dataset with different component proportions $\sigma$ in the Gym tasks. The datasets consist of $1 \times 10^6$ samples, where $\sigma$ of the samples originate from the *expert* dataset and the remaining $1 - \sigma$ of the samples come from the *random* dataset.

| Source | Task | $\sigma=0\%$ | $\sigma=10\%$ | $\sigma=20\%$ | $\sigma=30\%$ | $\sigma=40\%$ | $\sigma=50\%$ | $\sigma=100\%$ |
|---|---|---|---|---|---|---|---|---|
| medium expert | halfcheetah | $47.1_{\pm0.2}$ | $87.3_{\pm2.0}$ | $90.4_{\pm0.6}$ | $91.4_{\pm0.4}$ | $94.8_{\pm1.3}$ | $95.6_{\pm0.5}$ | $\mathbf{96.2}_{\pm0.3}$ |
| | hopper | $89.3_{\pm4.0}$ | $72.8_{\pm2.5}$ | $98.7_{\pm9.7}$ | $90.3_{\pm2.2}$ | $92.2_{\pm5.3}$ | $\mathbf{113.4}_{\pm0.5}$ | $111.7_{\pm0.4}$ |
| | walker2d | $84.9_{\pm0.2}$ | $79.4_{\pm3.9}$ | $95.2_{\pm9.3}$ | $94.7_{\pm5.3}$ | $111.5_{\pm2.3}$ | $112.1_{\pm0.2}$ | $\mathbf{112.4}_{\pm0.2}$ |
| random medium | halfcheetah | $25.6_{\pm1.0}$ | $\mathbf{48.5}_{\pm0.1}$ | $48.3_{\pm0.1}$ | $48.4_{\pm0.1}$ | $48.1_{\pm0.1}$ | $\mathbf{48.5}_{\pm0.3}$ | $47.1_{\pm0.2}$ |
| | hopper | $18.4_{\pm0.4}$ | $27.8_{\pm4.6}$ | $68.1_{\pm2.2}$ | $56.5_{\pm3.3}$ | $60.0_{\pm2.0}$ | $62.1_{\pm2.8}$ | $\mathbf{89.3}_{\pm4.0}$ |
| | walker2d | $3.6_{\pm1.7}$ | $36.0_{\pm11.9}$ | $77.7_{\pm1.2}$ | $72.3_{\pm4.2}$ | $74.1_{\pm4.2}$ | $82.3_{\pm0.4}$ | $\mathbf{84.9}_{\pm0.2}$ |
| random expert | halfcheetah | $25.6_{\pm1.0}$ | $81.0_{\pm1.2}$ | $86.2_{\pm0.1}$ | $90.6_{\pm0.3}$ | $90.7_{\pm0.1}$ | $90.3_{\pm1.6}$ | $\mathbf{96.2}_{\pm0.3}$ |
| | hopper | $18.4_{\pm0.4}$ | $40.3_{\pm6.9}$ | $77.0_{\pm7.3}$ | $108.7_{\pm0.7}$ | $111.1_{\pm0.7}$ | $\mathbf{112.5}_{\pm1.3}$ | $111.7_{\pm0.4}$ |
| | walker2d | $3.6_{\pm1.7}$ | $67.2_{\pm23.8}$ | $96.1_{\pm7.4}$ | $108.0_{\pm0.8}$ | $109.2_{\pm0.4}$ | $109.1_{\pm1.4}$ | $\mathbf{112.4}_{\pm0.2}$ |

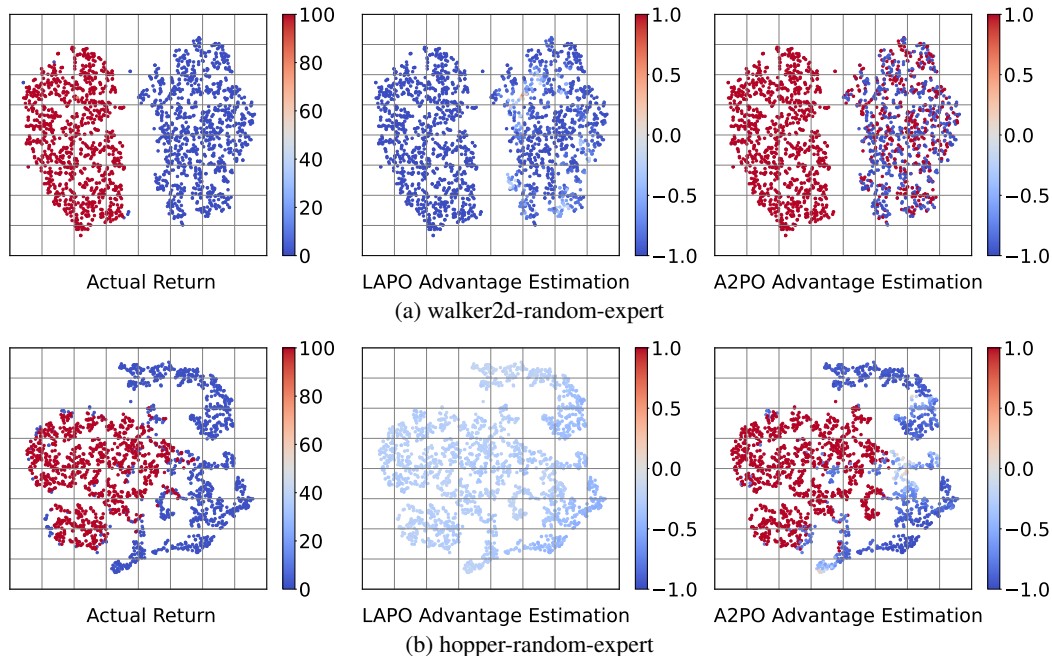

Figure 8: Comparison of our proposed A2PO method and the recent advanced AW method (LAPO) in advantage estimation for mixed-quality offline datasets (*random-expert*) in Gym tasks. Each data point represents an initial state-action pair in the offline dataset after applying PCA while varying shades of color indicate the magnitude of the actual return or advantage value.

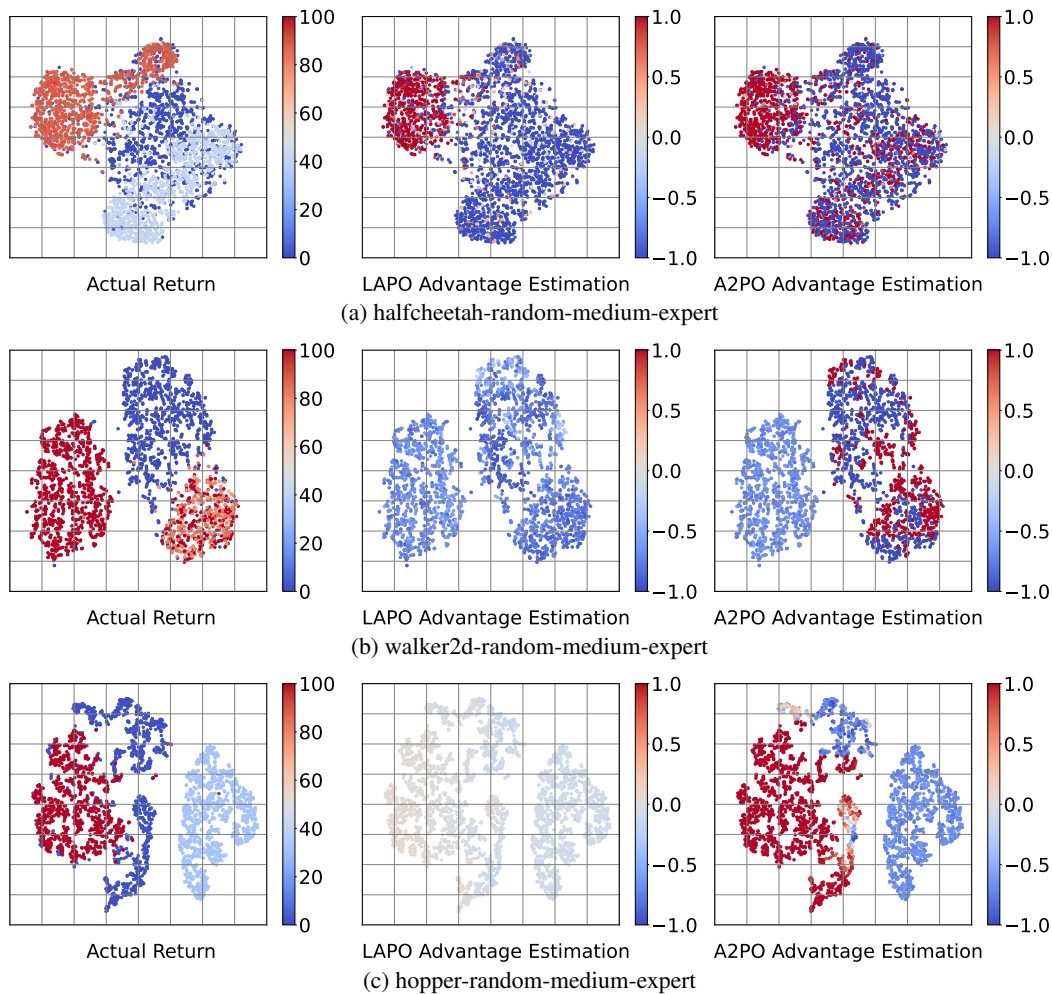

Figure 9: Comparison of our proposed A2PO method and the recent advanced AW method (LAPO) in advantage estimation for mixed-quality offline datasets (*random-medium-expert*) in locomotion tasks. Each data point represents an initial state-action pair in the offline dataset after applying PCA while varying shades of color indicate the magnitude of the actual return or advantage value.

