# OpenReview forum: "A2PO: Towards Effective Offline Reinforcement Learning from an Advantage-aware Perspective"
_NeurIPS.cc/2024/Conference — NeurIPS 2024 poster_

### Official Review · Reviewer_W3XG · 2024-07-11

**Soundness:** 3
**Presentation:** 3
**Contribution:** 3
**Rating:** 6
**Confidence:** 4

**Summary:**

This paper proposes an offline reinforcement learning method called A2PO, which aims to solve the problem of constraint conflicts in mixed-quality datasets collected from multiple behavior policies. A2PO optimizes offline learning by explicitly constructing advantage-aware policy constraints, especially in the presence of data of variable quality. Specifically, A2PO employs a CVAE to disentangle the distribution of actions under different behavior policies, modeling the advantage values of all training data as conditional variables. The agent can then optimize the advantage-aware policies based on these disentangled action distribution constraints to achieve high advantage values. Through extensive experiments on single- and mixed-quality datasets from the D4RL benchmark, A2PO demonstrates its superior performance over other offline RL baselines and advantage-weighted competitors.

**Strengths:**

1.	The authors introduce CVAE to deal with the problem of disentangling behavior policies in mixed-quality datasets, which is novel to me.
2.	The paper is well-structured and clearly written.
3.	Several analytical experiments are also provided to help understand the method

**Weaknesses:**

See questions.

**Questions:**

1.	Although this paper presents a new approach, the comparison with existing techniques may not be in depth enough and lacks an analysis of why the algorithm works. In addition, could the authors further discuss the conditions under which A2PO may fail?
2.	This paper focuses on the analysis and comparison of advantage-weighted offline RL methods, ignoring recent work on policy constraints in the same relative as A2PO, such as PRDC: Policy Regularization with Dataset Constraint for Offline. It would be more convincing if the authors could add research and experimental comparisons of related work to demonstrate the advantages of A2PO among its relative methods (even SOTA methods).
3.	Regarding implementation details, how can we ensure a sufficient number of high-quality samples in the dataset to support training when selecting ξ = 1? Additionally, how can we guarantee the accuracy of the introduced advantage information?

In general, this work demonstrates commendable quality and I would be willing to raise my score if my concerns could be addressed.

---

> ### Author Rebuttal · Authors · 2024-08-07
>
> Thanks for your support on the idea, writing, and experiment. And we will solve your doubts with text description and experimental support.
>
>   **[Q1: Although this paper presents a new approach, the comparison with existing techniques may not be in depth enough and lacks an analysis of why the algorithm works.]**
>
>   Sorry for the confusion.   In our original manuscript, we have compared our A2PO with other advanced offline RL methods. These techniques include policy constraint methods such as TD3+BC, value regularization methods like CQL, model-based method MOPO, diffusion-based method Diffusion-QL, and advantage-weighted methods like LAPO. Moreover, we have additionally conducted experiments to evaluate the return conditioned methods DT and %BC [1], as well as the data rebalancing techniques ReD [2] and its variation DeReD, as presented in Table R1. To validate the efficacy of the A2PO components, we have also conducted various ablation studies in the original paper. And we have the following key observations:
>
>   1. In Figure 1 of the paper, the toy demo showcases the erroneous constraint issue of the AW method and highlights the precise behavior distribution modeling capability of our A2PO, while the following comparing experiments further confirmed this point.
>   2. In comparison to conventional offline rl methods like TD3+BC/EQL and return-conditioned methods such as DT/%BC, the superiority of our A2PO becomes increasingly apparent as the substantial gaps between behavior policies become larger from medium, medium-expert to random-expert, random-expert-medium.
>   3. Although diffusion-based method Diffusion-QL precisely models the action distribution and gets comparable performance in several scenarios, our A2PO outperforms Diffusion-QL in most cases, particularly on mixed-quality datasets while being less time-consuming, as shown in Figure 7 of the paper.
>   4. The ablation study on the CVAE and the advantage-aware policy constraint (Appendices E, F) demonstrates the effectiveness of advantage-aware policy optimization.
>
>   These key observations highlight the effectiveness of A2PO to capture high-quality interactions within the dataset for a reasonable advantage-aware policy constraint and policy optimization.  Unlike previous AW methods focusing on redistributing the dataset but inadvertently reduce data diversity, our A2PO precisely disentangles and models the mixed behavior policies with advantage-condition technique on CVAE, then the agent can follow such disentangled action distribution constraints to optimize the advantage-aware policy towards high advantage values, therefore consistently improving the performance.
>
>   **[Q2: In addition, could the authors further discuss the conditions under which A2PO may fail?]**
>
>   Thank you for the constructive comments. The results of the original paper indicate that our A2PO does not perform well when the offline dataset distribution is narrow, such as halfcheetah-medium. When the dataset distribution is severely restricted, A2PO struggles to efficiently train both the CVAE and the agent model, which can result in task failure.
>
>   **[Q3: This paper focuses on the analysis and comparison of advantage-weighted offline RL methods, ignoring recent work on policy constraints in the same relative as A2PO, such as PRDC: Policy Regularization with Dataset Constraint for Offline Reinforcement Learning.]**
>
>   Thanks for the insight suggestion! We have additionally conducted experiments to compare the recently proposed policy-constrained method, PRDC [3] with our A2PO The results are presented in Table R1, which show that our A2PO consistently achieves superior performance on the majority of the gym, maze, antmaze, and adroit tasks. This comparison clearly underscores the superior performance and effectiveness of advantage-aware policy constraints employed in A2PO for offline policy optimization.
>
>
> **[Q4: Regarding implementation details, how can we ensure a sufficient number of high-quality samples in the dataset to support training when selecting ξ = 1? ]**
>
>   It is challenging to ensure that the dataset comprises an adequate number of high-quality samples. We address this limitation by utilizing the generalization capabilities of the generative model CVAE. By selecting $\xi=1$, the advantage-aware CVAE can generate and infer the optimal actions. As indicated in Table 6 of Appendix E, by directly inputting $\xi=1$, the CVAE-generated action $a\sim p_\psi(\cdot|z\sim\mathcal N(0,I),c)$ can also achieve high performance in various scenarios. Furthermore, we have also investigated the scenario where the dataset contains only a small quantity of high-quality samples, as shown in Appendix H. The results demonstrate that our A2PO algorithm is capable of achieving expert-level performance, even with a limited number of high-quality samples. These findings underscore the generalization ability of the advantage-aware CVAE and the robustness of our A2PO algorithm in deriving optimal policies across diverse structures of offline datasets.
>
> **[ Q5: How can we guarantee the accuracy of the introduced advantage information? ]**
>
>   Thanks for the insightful comments. The advantage value is determined by both the critic and the advantage estimation method. It is important to clarify that we do not propose a novel advantage computation method; rather, we adopt the widely accepted baseline in offline reinforcement learning, TD3+BC [4], as our framework. We utilize the advantage definition $ A(s, a) = Q(s, a) - V(s) $ for advantage estimation and subsequent normalization, ensuring that the agent maintains an effective estimation.
>
> [1] Decision Transformer: Reinforcement Learning via Sequence Modeling. NeurIPS 2021
>
> [2] Boosting Offline Reinforcement Learning via Data Rebalancing. arXiv 2022
>
> [3] Policy Regularization with Dataset Constraint for Offline Reinforcement Learning. ICML 2023
>
> [4] A Minimalist Approach to Offline Reinforcement Learning. NeurIPS 2021

---

> > ### Author Response · Authors · 2024-08-10
> >
> > Thank you for your review and comments. We hope that our additional evaluations and rebuttal have addressed your primary concerns with our paper. We would really appreciate feedback as to whether there are any (existing or new) points we have not covered, and we would be happy to address/discuss them!

---

> > > ### Comment · Reviewer_W3XG · 2024-08-11
> > >
> > > Thanks for your reply! I'm happy to see my concerns are well addressed. I have raised my score to 6.

---

> > > > ### Author Response · Authors · 2024-08-12
> > > >
> > > > We sincerely appreciate your positive feedback, the time and effort you have put into reviewing our responses and clarification. It is immensely gratifying to learn that our clarifications have helped address your main concerns.
> > > >
> > > > Thank you once again for your constructive feedback and support.

---

> ### Author Response · Authors · 2024-08-07
> **Experiment results for Q1 and Q3**
>
> Table R1. Test returns of our A2PO and return-conditioned and data rebalancing baselines. **Bold** indicates the best performance among the two algorithms. *Italic* indicates that the score are directly taken from the original paper. For the newly-constructed dataset, we rerun %BC, DT with the official code from [1]. As ReD[2] is a method without publicly available source code, we reimplement it ourselves.
>
>
>
> | Env                              | %BC           | DT            | CQL+ReD    | TD3BC+ReD  | IQL+DeReD stage 1 | IQL+DeReD stage 2 | IQL+ReD       | A2PO              |
> | -------------------------------- | ------------- | ------------- | ---------- | ---------- | ----------------- | ----------------- | ------------- | ----------------- |
> | halfcheetah-medium               | *42.5*        | *42.6*        | 48.2       | ***48.5*** | *47.5*            | *47.6*            | *47.6*        | 47.1$\pm$0.2      |
> | hopper-medium                    | *56.9*        | *67.6*        | *69.4*     | *59.3*     | *65.5*            | *65.1*            | *66.0*        | **80.3**$\pm$4.0  |
> | walker2d-medium                  | *75.0*        | *74.0*        | *83.5*     | *83.7*     | *74.5*            | *81.9*            | *78.6*        | **84.9**$\pm$0.2  |
> | Halfcheetah-medium-replay        | *40.6*        | *36.6*        | ***46.3*** | *44.7*     | *43.9*            | *43.4*            | *44.3*        | 44.8$\pm$0.2      |
> | hopper-medium-replay| *75.9*| *82.7*        | *98.6*     | *77.4*     | *92.8*            | *100.1*           | *101.0*       | **101.6**$\pm$1.3 |
> | walker2d-medium-replay| *62.5*| *66.6*        | ***86.7*** | *82.3*     | *72.9*            | *77.0*            | *79.5*        | 82.8$\pm$1.7      |
> | Halfcheetah-medium-expert| *92.9*        | *86.8*        | *81.6*     | *93.2*     | *87.9*| *91.8*| *92.6*| **95.6**$\pm$0.5  |
> | hopper-medium-expert| *110.9*       | *107.6*       | *95.0*     | *106.2*    | *89.3*| *104.7*| *106.1*| **113.4**$\pm$0.5 |
> | walker2d-medium-expert| *109.0*       | *108.1*       | *110.0*    | *110.0*    | *110.1*| *110.5*| *110.5*| **112.1**$\pm$0.2 |
> | halfcheetah-random-medium| 40.1$\pm$0.3  | 42.0$\pm$1.3  | -| -| -| -| 42.0$\pm$3.0  | **48.5**$\pm$0.3  |
> | hopper-random-medium| 21.0$\pm$5.9  | 3.1$\pm$0.0   | -| -| -| -| 6.7$\pm$2.9   | **62.1**$\pm$2.8  |
> | walker2d-random-medium| 32.0$\pm$8.1  | 66.9$\pm$8.4  | -| -| -| -| 60.0$\pm$2.6  | **82.3**$\pm$0.4  |
> | halfcheetah-random-expert| 7.7$\pm$2.7   | 10.3$\pm$7.4  | -| -| -| -| 42.4$\pm$26.1 | **90.3**$\pm$1.6  |
> | hopper-random-expert| 2.4$\pm$0.1   | 90.2$\pm$8.5  | -| -| -| -| 16.7$\pm$6.4  | **112.5**$\pm$1.3 |
> | walker2d-random-expert| 53.4$\pm$14.5 | 103.4$\pm$7.9 | -| -| -| -| 93.2$\pm$29.1 | **109.1**$\pm$1.4 |
> | halfcheetah-random-medium-expert | 29.1$\pm$5.2  | 42.6$\pm$0.6  | -| -| -| -| 39.1$\pm$21.6 | **90.6**$\pm$1.6  |
> | hopper-random-medium-expert      | 62.0$\pm$18.3 | 46.1$\pm$1.9  | -| -| -| -| 31.3$\pm$6.6  | **107.8**$\pm$0.4 |
> | walker2d-random-medium-expert    | 10.6$\pm$4.1  | 78.8$\pm$3.0  | -| -| -| -| 52.0$\pm$10.9 | **97.7**$\pm$6.7  |
>
> Table R2. Test returns of recent policy constraints Offline RL method PRDC and our A2PO.
>
> | Env| PRDC| A2PO|
> | -------------------------------- | ------------------- | ------------------ |
> | halfcheetah-medium| ***63.5**$\pm$0.9*  | 47.1$\pm$0.2       |
> | hopper-medium| ***100.3**$\pm$0.2* | 80.3$\pm$4.0       |
> | walker2d-medium| ***85.2**$\pm$0.4*  | **84.9**$\pm$0.2   |
> | Halfcheetah-medium-replay| ***55.0**$\pm$1.1*  | 44.8$\pm$0.2       |
> | hopper-medium-replay| *100.1$\pm$1.6*     | **101.6**$\pm$1.3  |
> | walker2d-medium-replay| ***92.0**$\pm$1.6*  | 82.8$\pm$1.7       |
> | Halfcheetah-medium-expert| *94.5$\pm$0.5*      | **95.6**$\pm$0.5   |
> | hopper-medium-expert| *109.2$\pm$4.0*     | **113.4**$\pm$0.5  |
> | walker2d-medium-expert| *111.2$\pm$0.6*     | **112.1**$\pm$0.2  |
> | halfcheetah-random-medium| **56.5**$\pm$2.6    | 48.5$\pm$0.3       |
> | hopper-random-medium| 5.5$\pm$0.4         | **62.1**$\pm$2.8   |
> | walker2d-random-medium| 5.5$\pm$0.8         | **82.3**$\pm$0.4   |
> | halfcheetah-random-expert| 1.3$\pm$0.5         | **90.3**$\pm$1.6   |
> | hopper-random-expert| 24.8$\pm$14.6       | **112.5**$\pm$1.3  |
> | walker2d-random-expert| 1.1$\pm$0.7         | **109.1**$\pm$1.4  |
> | halfcheetah-random-medium-expert | 10.5$\pm$2.8        | **90.6**$\pm$1.6   |
> | hopper-random-medium-expert| 88.5$\pm$15.2| **107.8**$\pm$0.4  |
> | walker2d-random-medium-expert| 4.87$\pm$3.2| **97.7**$\pm$6.7   |
> | maze2d-umaze| 127.4$\pm$23.4| **133.3**$\pm$9.6  |
> | maze2d-medium| 60.0$\pm$5.3| **114.9**$\pm$12.9 |
> | maze2d-large| 151.6$\pm$16.0| **156.4**$\pm$5.8  |
> | antmaze-umaze-diverse-v2| *90.0$\pm$6.8*      | **93.3**$\pm$4.7   |
> | antmaze-medium-diverse-v2| *78.8$\pm$6.9*      | **86.7**$\pm$9.4   |
> | antmaze-large-diverse-v2| *50.0$\pm$5.4*      | **53.3**$\pm$4,7   |

---

### Official Review · Reviewer_M64c · 2024-07-12

**Soundness:** 3
**Presentation:** 3
**Contribution:** 3
**Rating:** 7
**Confidence:** 3

**Summary:**

The authors propose an advantage aware offline RL algorithm for datasets consisting of data from multiple behavior policies. The algorithm consists of two steps, (1) behavior policy disentangling: Wherein a CVAE is trained to output actions conditioned on normalized advantage and state. (2) Policy optimization, where the policy is optimized based on the actions generated by the CVAE. The authors demonstrate the performance of the algorithms on standard benchmarks

**Strengths:**

1. The paper is well written and easy to follow

2. The experimental section is extensive

**Weaknesses:**

1. **The Agent policy optimization is unclear**

    1. Why is a behavior regularization term (2nd term in (8)) needed when you have a generative model trained on the dataset?

    2. Why not condition the 2nd in eq 8 term on c*?

2. **Missing some return conditioned baselines**

    1. Comparison with return conditioned approaches such as %BC and Decision Transformers are needed to understand the advantage of the proposed approach

    2. Comparisons with data rebalancing techniques [1] can help understand methods with mixed datasets

[1] Yue, Yang, et al. "Boosting offline reinforcement learning via data rebalancing." arXiv preprint arXiv:2210.09241 (2022).

**Questions:**

See weaknesses

**Limitations:**

The authors address limitations

---

> ### Author Rebuttal · Authors · 2024-08-07
>
> Thanks for your positive comments on the experiment's completeness and the writing's quality. We will address your concerns in this section.
>
> **[Q1.1: The Agent policy optimization is unclear: Why is a behavior regularization term (2nd term in (8)) needed when you have a generative model trained on the dataset?]**
>
> Sorry for the confusion.  First of all,  as an offline RL method with policy constraints, A2PO requires the agent policy to closely align with the behavior policy. In order to achieve this, we incorporate a behavior regularization term that constrains the action distribution of the advantage-aware agent based on the computed advantage value.
>
> Meanwhile, although latent action space inherently imposes a constraint on the action output [1], the regularization term ensures that the actions chosen by the A2PO policy align with the advantage condition $\xi$ determined by the critic, thereby providing a more precise policy constraint.   To confirm our perspective, we conducted an ablation study by eliminating the regularization term in Table 7 and Appendix F of the original paper. The results of this study demonstrate that the regularization term in A2PO enhances its performance in most cases, providing strong support for the necessity and effectiveness of the regularization term.
>
> **[Q1.2: The Agent policy optimization is unclear: Why not condition the 2nd in eg 8 term on *c*?]**
>
> Thanks for your insightful comment. By directly constraining the agent's output on $c^*$, the agent will become fully constrained on the 'optimal' action generated by the CVAE. However, this approach may introduce additional computing errors due to the training of the CVAE. Instead, we propose utilizing the critic-computed $c$ and constraining the agent on the precisely sampled action. This not only avoids errors but also allows the agent to be aware of the multiple action distributions of behavior policies within the datasets.
>
> To further verify our statement, we conducted an ablation study on the 'random-medium', 'random-expert' and 'random-medium' dataset by changing the regularization term from the original advantage-aware form $\mathbb E_{\substack{(s,a)\sim \mathcal D, \tilde{z}\sim \pi_\omega(\cdot|c), \\ a_\xi\sim p_\psi(\cdot|\tilde{z}, c)}}\big[(a-a_\xi)^2\big]$ to $\mathbb E_{\substack{(s,a)\sim \mathcal D, \tilde{z}^*\sim \pi_\omega(\cdot|c^*), \\ a_\xi^*\sim p_\psi(\cdot|\tilde{z}^*, c^*),\\ a_\text{cvae}^*\sim p_\psi(\cdot|z, c^*), z\sim \mathcal N(0,I)}}\big[(a_\xi^*-a_\text{cvae}^*)^2\big]$.  Therefore, the output of the A2PO optimal policy $a_\xi^*$ is constrained only by the CVAE-inferred best action $a_\text{cvae}^*$.. The results are presented in Table R1， which show that our original form outperforms the variation in most cases, providing support for our regularization term.
>
> Table R1. Test returns of A2PO and A2PO constrined on $a_\text{cvae}^*\sim p_\psi(\cdot|z=\mathcal N(0,I), c^*)$. **Bold** indicates the best performance among the two algorithms.
>
> | Env                              | A2PO constrined on  $a_\text{cvae}^*$ | A2PO              |
> | -------------------------------- | ------------------------------------- | ----------------- |
> | halfcheetah-random-medium        | 41.0$\pm$1.6                          | 48.5$\pm$0.3      |
> | hopper-random-medium             | 40.9$\pm$2.0                          | **62.1**$\pm$2.8  |
> | walker2d-random-medium           | 57.8$\pm$3.4                          | **82.3**$\pm$0.4  |
> | halfcheetah-random-expert        | **93.1**$\pm$6.2                      | 90.3$\pm$1.6      |
> | hopper-random-expert             | 81.8$\pm$0.2                          | **112.5**$\pm$1.3 |
> | walker2d-random-expert           | 96.8$\pm$3.3                          | **109.1**$\pm$1.4 |
> | halfcheetah-random-medium-expert | 89.0$\pm$4.8                          | **90.6**$\pm$1.6  |
> | hopper-random-medium-expert      | 12.8$\pm$4.0                          | **107.8**$\pm$0.4 |
> | walker2d-random-medium-expert    | 63.0$\pm$4.5                          | **97.7**$\pm$6.7  |
>
> **[Q2: Missing some return conditioned baselines：1. Comparison with return conditioned approaches such as %BC and Decision Transformers are needed to understand the advantage of the proposed approach;  2.Comparisons with data rebalancing techniques can help understand methods with mixed datasets]**
>
> Thanks for your constructive suggestions! We have additionally compared our A2PO with the return condition baselines 10%BC and Decision Transformer [2], and Offline RL with data rebalancing technique ReD and its variation DeReD[3]. The results are presented in Table R2. We directly take the score from the original paper for 'medium', 'medium-replay' and 'medium-expert' datasets.  For the new 'random-medium',  'random-expert' and 'random-medium-expert' datasets, due to the time limit, we only evaluate IQL with ReD. The results show that our A2PO outperforms both the return condition baselines and data rebalancing techniques in most cases, which further demonstrates the superiority of our advantage-aware methods to tackle the constrain conflict issue.
>
>
> [1] PLAS: Latent Action Space for Offline Reinforcement Learning. CoRL 2021
>
> [2] Decision Transformer: Reinforcement Learning via Sequence Modeling. NeurIPS 2021
>
> [3] Boosting Offline Reinforcement Learning via Data Rebalancing. arXiv 2022

---

> ### Author Response · Authors · 2024-08-07
> **Experiment results for Q2**
>
> Table R2. Test returns of our A2PO and other return conditioned and data rebalancing baselines. *Italic* indicates that the scores are directly taken from the original paper. For the newly constructed dataset, we rerun %BC and DT with the official code from [2]. As ReD[3] is a method without publicly available source code, we reimplement it ourselves.
>
> | Env                              | %BC           | DT            | CQL+ReD    | TD3BC+ReD  | IQL+DeReD stage 1 | IQL+DeReD stage 2 | IQL+ReD       | A2PO              |
> | -------------------------------- | ------------- | ------------- | ---------- | ---------- | ----------------- | ----------------- | ------------- | ----------------- |
> | halfcheetah-medium               | *42.5*        | *42.6*        | 48.2       | ***48.5*** | *47.5*            | *47.6*            | *47.6*        | 47.1$\pm$0.2      |
> | hopper-medium                    | *56.9*        | *67.6*        | *69.4*     | *59.3*     | *65.5*            | *65.1*            | *66.0*        | **80.3**$\pm$4.0  |
> | walker2d-medium                  | *75.0*        | *74.0*        | *83.5*     | *83.7*     | *74.5*            | *81.9*            | *78.6*        | **84.9**$\pm$0.2  |
> | Halfcheetah-medium-replay        | *40.6*        | *36.6*        | ***46.3*** | *44.7*     | *43.9*            | *43.4*            | *44.3*        | 44.8$\pm$0.2      |
> | hopper-medium-replay             | *75.9*        | *82.7*        | *98.6*     | *77.4*     | *92.8*            | *100.1*           | *101.0*       | **101.6**$\pm$1.3 |
> | walker2d-medium-replay           | *62.5*        | *66.6*        | ***86.7*** | *82.3*     | *72.9*            | *77.0*            | *79.5*        | 82.8$\pm$1.7      |
> | Halfcheetah-medium-expert        | *92.9*        | *86.8*        | *81.6*     | *93.2*     | *87.9*            | *91.8*            | *92.6*        | **95.6**$\pm$0.5  |
> | hopper-medium-expert             | *110.9*       | *107.6*       | *95.0*     | *106.2*    | *89.3*            | *104.7*           | *106.1*       | **113.4**$\pm$0.5 |
> | walker2d-medium-expert           | *109.0*       | *108.1*       | *110.0*    | *110.0*    | *110.1*           | *110.5*           | *110.5*       | **112.1**$\pm$0.2 |
> | halfcheetah-random-medium        | 40.1$\pm$0.3  | 42.0$\pm$1.3  | -          | -          | -                 | -                 | 42.0$\pm$3.0  | **48.5**$\pm$0.3  |
> | hopper-random-medium             | 21.0$\pm$5.9  | 3.1$\pm$0.0   | -          | -          | -                 | -                 | 6.7$\pm$2.9   | **62.1**$\pm$2.8  |
> | walker2d-random-medium           | 32.0$\pm$8.1  | 66.9$\pm$8.4  | -          | -          | -                 | -                 | 60.0$\pm$2.6  | **82.3**$\pm$0.4  |
> | halfcheetah-random-expert        | 7.7$\pm$2.7   | 10.3$\pm$7.4  | -          | -          | -                 | -                 | 42.4$\pm$26.1 | **90.3**$\pm$1.6  |
> | hopper-random-expert             | 2.4$\pm$0.1   | 90.2$\pm$8.5  | -          | -          | -                 | -                 | 16.7$\pm$6.4  | **112.5**$\pm$1.3 |
> | walker2d-random-expert           | 53.4$\pm$14.5 | 103.4$\pm$7.9 | -          | -          | -                 | -                 | 93.2$\pm$29.1 | **109.1**$\pm$1.4 |
> | halfcheetah-random-medium-expert | 29.1$\pm$5.2  | 42.6$\pm$0.6  | -          | -          | -                 | -                 | 39.1$\pm$21.6 | **90.6**$\pm$1.6  |
> | hopper-random-medium-expert      | 62.0$\pm$18.3 | 46.1$\pm$1.9  | -          | -          | -                 | -                 | 31.3$\pm$6.6  | **107.8**$\pm$0.4 |
> | walker2d-random-medium-expert    | 10.6$\pm$4.1  | 78.8$\pm$3.0  | -          | -          | -                 | -                 | 52.0$\pm$10.9 | **97.7**$\pm$6.7  |

---

> > ### Comment · Reviewer_M64c · 2024-08-07
> >
> > I would like to thank the authors for their detailed response. I have updated my score accordingly. I wish the authors the best!

---

> > > ### Author Response · Authors · 2024-08-10
> > >
> > > We sincerely appreciate your positive feedback, the time and effort you have put into reviewing our responses and clarification. It is immensely gratifying to learn that our clarifications have helped address your main concerns.
> > >
> > > Thank you once again for your constructive feedback and support.

---

### Official Review · Reviewer_xTq4 · 2024-07-12

**Soundness:** 3
**Presentation:** 3
**Contribution:** 2
**Rating:** 6
**Confidence:** 4

**Summary:**

The paper proposes using CVAE to train an agent and using a decoder, which conditions on state and advantage value, to generate actions. The Q function and V function are trained simultaneously, and the policy is trained in a TD3-BC style. Experiments show that the method can outperform baselines in most tasks.

**Strengths:**

The paper is easy to follow, and the motivation and intuition for the algorithm design are reasonable. The experiments are comprehensive and justify the advantages of the method claimed by the authors. Conditioning on the advantage value, instead of using it as a coefficient, is interesting, allowing for sampling to make the advantage value extreme ($\xi=1$) to get the optimal action.

**Weaknesses:**

I found some results in the experimental section problematic. More details are in the Questions section.

**Questions:**

1. In line 169, what prior distribution of $p(z)$ did you choose?

2. When training the Q and V functions (Eq(7)), the authors claimed that training simultaneously can stabilize training. Do the authors have any intuition, theory, or literature to back up such a design? I ask because, in my memory, most methods that need to estimate the V function train Q and V separately, such as IQL[1] and SAC[2]. Simultaneous training may cause instability, similar to training GANs.

3. In Table 1, I found some baseline results inconsistent with the original paper. For example, hopper-medium-v2 is 90.5 for Diffusion-QL in the original paper, but 70.3 in this paper; antmaze-umaze-diverse is 66.2 for Diffusion-QL in the original paper, but 24 in this paper. Most of the results for Diffusion-QL are not consistent with the original paper, while in line 234, the authors claim their results are from the original paper. Can the authors explain the mismatch?

4. In line 463, the authors state they use the v2 version for antmaze tasks, while most other baselines, including IQL and Diffusion-QL, use the v0 version. In my experience, the v2 version dataset can always provide a higher score than the v0 version when using the same algorithm. It is unfair to compare A2PO's v2 results with the v0 results of other baselines.

5. I found that the training curve of hopper random-medium-expert in Figure 4 is unstable. Can the authors provide more training curves for other environments?


[1] Kostrikov, Ilya, Ashvin Nair, and Sergey Levine. "Offline reinforcement learning with implicit q-learning." arXiv preprint arXiv:2110.06169 (2021).

[2]Haarnoja, Tuomas, et al. "Soft actor-critic algorithms and applications." arXiv preprint arXiv:1812.05905 (2018).

**Limitations:**

My main concern is about the empirical experiments. It seems the algorithm doesn't have competitive performance compared to diffusion-based policy algorithms.

---

> ### Author Rebuttal · Authors · 2024-08-07
>
> Thank you for affirming the idea's novelty and the experiment's comprehensiveness. In this section, we aim to address your concerns.
>
> **[Q1: In line 169, what prior distribution of $p(z)$ did you choose?]**
>
> Sorry for the confusion. As stated in line 167 of the original paper, the prior distribution of $p(z)$ is chosen to be $\mathcal N(0,I)$.
>
> **[Q2: When training the Q and V functions (Eq(7)), the authors claimed that training simultaneously can stabilize training. Do the authors have any intuition, theory, or literature to back up such a design?]**
>
> Thank you for your insightful question. We apologize for the confusion caused by the misdescription in the manuscript. In fact, we did not train the Q and V functions simultaneously; instead, we employed an alternating training approach for policy evaluation following the LAPO method [1].  Specifically,  the Q-function is still optimized with the Bellman equation: $L_Q(\theta)=\mathbb E_{(s,a,r,s')\sim \mathcal D}\sum_i [r+\gamma V_{\hat \phi}(s')-Q_{\theta_i}(s,a)]^2$, while the V-function is optimized with the the expectation on Q-value: $L_V(\phi)=\mathbb E_{s\sim \mathcal D}[V_\phi(s)-\mathbb E_{ \tilde z^*\sim \pi_\omega (\cdot|c^*), a^*_\xi\sim p_\psi(\cdot|\tilde z^*, c^*) }Q_\hat\theta(s,a_\xi^*)]^2$. Corresponding to the implementation presented in the supplementary material, in line 230-238 of`latent_a.py`, the V-function update codes are:
>
> ```python
> adv_sample = torch.ones((next_s.size()[0], 1)).to(self.device)        #get \xi^*
> latent_a = self.actor_target(torch.cat((s, adv_sample), dim=1))	      #get \tilde z^*
> hat_a = self.vae.decode(torch.cat((s, adv_sample), dim=1), latent_a)  #get a_\xi^*
> temp_q1, temp_q2 = self.critic_target(s,hat_a)                        #get Q value
> target_v = torch.min(temp_q1, temp_q1) * self.doubleq_min + torch.max(temp_q1, temp_q1) * (1 - self.doubleq_min)
> current_v = self.critic.v(s)
> loss = ... + F.mse_loss(current_v, target_v)
> ```
>
>
>
> **[Q3: In Table 1, I found some baseline results inconsistent with the original paper.]**
>
> Sorry for the confusion. We have systematically reviewed the score tables for the 'medium', 'medium-replay', and 'medium-expert' datasets, rectifying any recording errors identified. The mismatch scores reported in the original manuscript were obtained by re-running the corresponding source code, which was accidently mixed with the officially reported scores.
> We give a new comparison table between these baselines and A2PO, as illustrated in Table R1, and we will update the paper accordingly.
>
> **[Q4: It is unfair to compare A2PO's v2 results with the v0 results of other baselines.]**
>
> Thanks for your constructive comments. We have additionally tested A2PO using the v0 version for the AntMaze tasks. For BC, BCQ, TD3+BC, CQL, AWAC, IQL, and Diffusion-QL, we have directly referenced the scores from the Diffusion-QL paper [1]. For CQL+AW, we have utilized the scores from the CQL+AW paper [2]. For EQL and LAPO—neither of which provide scores for the v0 version—we have re-evaluated their performance, presenting both the scores and standard deviations. The results are summarized in Table R2, indicating that our A2PO continues to achieve performance comparable to that of advanced baselines such as Diffusion-QL.
>
> **[Q5: I found that the training curve of hopper random-medium-expert in Figure 4 is unstable. Can the authors provide more training curves for other environments. ]**
>
> Thanks for the insightful comment. We have provided various A2PO training curves in the supplementary pdf, in which the A2PO is trained with 1M timesteps while performing online evaluation at each 2K timesteps. The results show that the A2PO maintains overall stability during training on most of the tasks and datasets.
>
> **[Q6: My main concern is about the empirical experiments. It seems the algorithm doesn't have competitive performance compared to diffusion-based policy algorithms.]**
>
> We appreciate your valuable comment; however, we respectfully disagree with the assertion that A2PO does not exhibit competitive performance compared to diffusion policy for the following reasons:
>
> 1. Even taking into account the comparative data from the Diffusion-QL paper, our A2PO method still outperforms diffusion-QL in 11 out of 18 gym tasks, 4 out of 6 maze tasks, 2 out of 3 kitchen tasks, and 1 out of 2 adroit tasks. This demonstrates that our algorithm exhibits a comparative advantage in most task scenarios over the diffusion policy.
> 2. The primary focus of our research is not on benchmarking against diffusion-based methods but on addressing the conflict constraint issue, which current AW methods fail to solve effectively. Our extensive experiments indicate that A2PO significantly outperforms existing AW methods, such as LAPO.
> 3. Our approach is orthogonal to diffusion-based methods. Specifically, Diffusion-QL utilizes diffusion as the policy to implicitly capture the multimodal action distribution, while it overlooks the intrinsic quality of different actions. In contrast, our A2PO explicitly leverages the advantage value to disentangle the action distributions of interrelated behavior policies, ensuring that the quality of different actions is positively correlated with the advantage value. Therefore, our advantage-aware mechanism can be effectively integrated with diffusion-based methods, enhancing their capacity to identify multimodal data of varying quality.
>
>  In conclusion, our core contribution lies not in the model itself but in the introduction of an advantage-aware policy.
>
> [1] Latent-Variable Advantage-Weighted Policy Optimization for Offline RL. NeurIPS 2022
>
> [2] Diffusion Policies as an Expressive Policy Class for Offline Reinforcement Learning. ICLR 2023
>
> [3] Harnessing Mixed Offline Reinforcement Learning Datasets via Trajectory Weighting. ICLR 2023

---

> > ### Author Response · Authors · 2024-08-10
> >
> > Thank you for your review and comments. We hope that our additional evaluations and rebuttal have effectively addressed your primary concerns regarding our paper. We would greatly appreciate any feedback on whether there are existing or new points that we have not yet covered, and we would be glad to address or discuss them further.

---

> > > ### Comment · Reviewer_xTq4 · 2024-08-10
> > >
> > > Thank you for the authors' detailed response. My concerns have been addressed, and as a result, I have increased my score to 6. I do have a final question for discussion, which will not affect my score.
> > >
> > > Why is an additional actor $\pi$ needed to learn the latent variable $\tilde{z}^*$ (Eq 8)? What would happen if we simply input a random vector instead? Since $\xi$ is conditioned during the training of the CVAE, I wonder if directly inputting $\xi^*$ and a random $z$ would also produce a high Q-value action.

---

> > > > ### Author Response · Authors · 2024-08-12
> > > >
> > > > We sincerely appreciate your positive feedback, the time and effort you have put into reviewing our responses and clarification. We will address your doubt here.
> > > >
> > > > **[Q1: Why is an additional actor π needed to learn the latent variable z~∗ (Eq 8)? What would happen if we simply input a random vector instead?]**
> > > >
> > > > The random latent-input policy mentioned is the CVAE policy outlined in our manuscript. Specifically, the CVAE policy directly takes the state-advantage  $c^*=s||\xi^*$ and random noise $z_0 \sim \mathcal{N}(0,I)$ as input to the CVAE decoder to get action output. Although the advantage-aware CVAE,can effectively generate actions that adhere to the corresponding in-sample distribution, the A2PO actor $\pi_\omega$ provides a more precise mechanism for action selection by directly outputting a discrete $\tilde{z}^*$ within the latent space.
> > > >
> > > > Meanwhile, our experimental findings, detailed in Appendix E, demonstrate a consistent performance advantage of the A2PO policy over the CVAE policy across the majority of scenarios. This performance disparity is particularly pronounced in the newly constructed mixed-quality datasets, such as the random-expert and random-medium-expert scenarios. These results indicate that A2PO successfully achieves well-disentangled behavior policies and an optimal agent policy, thereby exceeding the capabilities of CVAE-reconstructed behavior policies.

---

> ### Author Response · Authors · 2024-08-07
> **Experiment results for Q3 and Q4**
>
> Table R1. Test returns of A2PO and other baselines in gym tasks. **Bold** indicates the best performance among these algorithms.
>
> |                           | CQL     | EQL     | Diffusion-QL | CQL+AW | A2PO              |
> | ------------------------- | ------- | ------- | ------------ | ------ | ----------------- |
> | halfcheetah-medium        | *44.0*  | *47.2*  | *51.1*       | *49*   | 47.1$\pm$0.2      |
> | hopper-medium             | *58.5*  | *70.6*  | ***90.5***   | *71*   | 80.3$\pm$4.0      |
> | walker2d-medium           | *72.5*  | *83.2*  | ***87.0***   | *83*   | 84.9$\pm$0.2      |
> | Halfcheetah-medium-replay | *45.5*  | *44.5*  | ***47.8***   | *47*   | 44.8$\pm$0.2      |
> | hopper-medium-replay      | *95.0*  | *98.1*  | *101.3*      | *99*   | **101.6**$\pm$1.3 |
> | walker2d-medium-replay    | *77.2*  | *81.6*  | ***95.5***   | *87*   | 82.8$\pm$1.7      |
> | Halfcheetah-medium-expert | *91.6*  | *94.6*  | ***96.8***   | *84*   | 95.6$\pm$0.5      |
> | hopper-medium-expert      | *105.4* | *111.5* | *111.1*      | *91*   | **113.4**$\pm$0.5 |
> | walker2d-medium-expert    | *108.8* | *110.2* | *110.1*      | *109*  | **112.1**$\pm$0.2 |
>
>
>
> Table R2. Test returns of A2PO and other baselines in antmaze v0 tasks.
>
> |                       Env | BC     | BCQ    | TD3+BC | CQL        | EQL           | Diffusion-QL | AWAC   | IQL    | CQL+AW | LAPO         | A2PO             |
> | ------------------------: | ------ | ------ | ------ | ---------- | ------------- | ------------ | ------ | ------ | ------ | ------------ | ---------------- |
> |  antmaze-umaze-diverse-v0 | *45.6* | *55.0* | *71.4* | ***84.0*** | 50.8$\pm$11.6 | *66.2*       | *49.3* | *62.2* | *54*   | 0.0$\pm$0.0  | 72.6$\pm$10.2    |
> | antmaze-medium-diverse-v0 | *0.0*  | *0.0*  | *3.0*  | *53.7*     | 62.2$\pm$6.7  | *78.6*       | *0.7*  | 70.0   | *24*   | 30.2$\pm$9.4 | **80.2**$\pm$4.0 |
> |  antmaze-large-diverse-v0 | *0.0*  | *2.2*  | *0.0*  | *14.9*     | 38.0$\pm$5.5  | ***56.6***   | *1.0*  | *47.5* | *40*   | 22.3$\pm$4.8 | 52.1$\pm$7.9     |

---

### Official Review · Reviewer_e3Zq · 2024-07-13

**Soundness:** 4
**Presentation:** 4
**Contribution:** 3
**Rating:** 8
**Confidence:** 3

**Summary:**

This paper presents A2PO. A2PO is an offline RL method for learning with datasets that was collected by a diverse set of poliices. The aim of the method is to disentangle the data being collected by each policy by using advantage calculation and then using this information to better learn policies from the dataset. Specifically, they use a VAE and condition it on advantage and state and then train in latent states. They perform a thorough set of experiments on benchmarks to compare to baseline methods. They also perform an ablation study to find the important qualities in their method.

**Strengths:**

Originality. Average strength. The originalty of this work is what I would expect. They have with a novel idea of disentangling the policies using advantage estimation which builds upon the previous idea of advantage estimation. I think this work is of solid novelty.

Quality. Major strength. The emperical analysis in this work is done very well. They compare to many baselines and show their method is clearly superior in many of the domains. They also perform a very good ablation study that shows the quality of their choises. I think this is a high point of the paper.

Clarity. Major strength. The paper is very well written, provides excellent figures and is easy to follow. Great job here.

Significance. Average strength. The work here seems like it takes a good step forward in the domain of offline RL with mixed quality datasets. The significance here is solid due to their impressive results but no more than I would expect from a good paper.

**Weaknesses:**

I don't have any major comments here. I believe this paper is complete, written well, has impressive results and is easy to follow.

**Questions:**

None.

**Limitations:**

The limitation they address is that the method is slower than others, but no slower than some baselines they provide. I believe this is addressed well enough and improving upon the computation speed is clearly out of scope.

---

> ### Author Rebuttal · Authors · 2024-08-07
>
> We sincerely appreciate your positive support of our work. In the future, we will make further improvements to the A2PO to contribute to the offline rl community.

---

### Official Review · Reviewer_3MzD · 2024-07-14

**Soundness:** 3
**Presentation:** 3
**Contribution:** 3
**Rating:** 7
**Confidence:** 3

**Summary:**

The paper introduces A2PO, a novel approach to offline reinforcement learning that addresses the constraint conflict issue in mixed-quality datasets by using a Conditional Variational Auto-Encoder to disentangle action distributions and optimize policies towards high advantage values. A2PO demonstrates superior performance over existing methods on the D4RL benchmark, showcasing its effectiveness in leveraging diverse offline datasets for more robust policy learning.

**Strengths:**

see questions part

**Weaknesses:**

see questions part

**Questions:**

This paper focuses on the constraint conflict issue in offline reinforcement learning when dealing with mixed-quality datasets collected from multiple behavior policies. By using a Conditional Variational Auto-Encoder (CVAE), A2PO disentangles the action distributions of different behavior policies present in the mixed-quality dataset. The CVAE models the advantage values as conditional variables, allowing it to generate action distributions that reflect the behavior policies' distinct characteristics.
In general, this paper is interesting and well-written. One question is that since we can directly select the good trajectories according to the rewards, is there any ablation study to show that CVAE can do better than this simple method?

**Limitations:**

see questions part

---

> ### Author Rebuttal · Authors · 2024-08-07
>
> Thanks for your supportive review of our method, experiments, and writing skills. We will address your doubts here.
>
> **[Q1. since we can directly select the good trajectories according to the rewards, is there any ablation study to show that CVAE can do better than this simple method?]**
>
>   Thanks for your constructive suggestions! We have additionally conducted ablation study on our CVAE in the A2PO method. To illustrate the effectiveness of our CVAE approach compared to merely selecting high-quality trajectories for training, we eliminate the CVAE module and implement the trajectory selection method proposed in [1,2], which involves selecting only the top 10% of trajectories based on their returns for agent training.
>
>   The results shown in Table R1 clearly indicate that A2PO without CVAE [top-10%] consistently lags behind the original A2PO in most cases, especially on the mixed-qualities datasets like 'walker2d-random-medium' and 'walker2d-random-expert'.   This is mainly due to the challenge of determining the hyperparameter $k$ for selecting the top-$k$ trajectories for training across different tasks and datasets. A smaller $k$ filters out more data, limiting the agent's awareness of the action distribution and task information. On the other hand, a larger $k$ introduces constraint conflicts that hinder agent policy improvement.   Instead, our A2PO utilizes CVAE conditioned on advantage value to obtain disentangled behavior constraints.  By optimizing the advantage-aware policy based on these disentangled action distribution constraints, we are able to achieve higher returns by directing the agent toward high advantage values.
>
>
>
>   Table R1. Test returns of A2PO and A2PO w/o CVAE [selecting top-10% trajectory for training]. **Bold** indicates the best performance among the two algorithms.
>
>   | Env                              | A2PO w/o CVAE [top-10%] | A2PO              |
>   | -------------------------------- | ----------------------- | ----------------- |
>   | halfcheetah-medium               | 42.1$\pm$0.1            | **47.1**$\pm$0.2  |
>   | hopper-medium                    | 61.2$\pm$9.0            | **80.3**$\pm$4.0  |
>   | walker2d-medium                  | 17.9$\pm$1.2            | **84.9**$\pm$0.2  |
>   | halfcheetah-medium-replay        | 35.6$\pm$2.9            | **44.8**$\pm$0.2  |
>   | hopper-medium-replay             | 89.8$\pm$1.1            | **101.6**$\pm$1.3 |
>   | walker2d-medium-replay           | 73.9$\pm$3.4            | **82.8**$\pm$1.7  |
>   | Halfcheetah-medium-expert        | 87.2$\pm$0.2            | **95.6**$\pm$0.5  |
>   | hopper-medium-expert             | 103.9$\pm$0.9           | **113.4**$\pm$0.5 |
>   | walker2d-medium-expert           | 111.1$\pm$0.1           | **112.1**$\pm$0.2 |
>   | halfcheetah-random-medium        | 45.9.$\pm$0.2           | **48.5**$\pm$0.3  |
>   | hopper-random-medium             | 60.1$\pm$1.5            | **62.1**$\pm$2.8  |
>   | walker2d-random-medium           | 3.2$\pm$1.8             | **82.3**$\pm$0.4  |
>   | halfcheetah-random-expert        | 54.1$\pm$2.1            | **90.3**$\pm$1.6  |
>   | hopper-random-expert             | 105.8$\pm$1.0           | **112.5**$\pm$1.3 |
>   | walker2d-random-expert           | 10.5$\pm$1.2            | **109.1**$\pm$1.4 |
>   | halfcheetah-random-medium-expert | 50.3.$\pm$1.0           | **90.6**$\pm$1.6  |
>   | hopper-random-medium-expert      | 101.3$\pm$3.9           | **107.8**$\pm$0.4 |
>   | walker2d-random-medium-expert    | 15.2$\pm$0.6            | **97.7**$\pm$6.7  |
>
>   [1] Decision Transformer: Reinforcement Learning via Sequence Modeling. NeurIPS 2021.
>
>   [2] Harnessing Mixed Offline Reinforcement Learning Datasets via Trajectory Weighting. ICLR 2023.

---

> > ### Comment · Reviewer_3MzD · 2024-08-14
> >
> > Thank you for the reply! My concern is solved and I would like to improve my rating to 7.

---

### Author Rebuttal · Authors · 2024-08-07

Please refer to the attachment for the figures of all results during rebuttal.

---

### Author Response · Authors · 2024-08-14

We thank the reviewers for the feedback!.
Since the discussion period is ending soon, we give a brief summarization here,
including the comments from the reviewers and the future revision according to the reviewers's opinion.

- Reviewer's comments
  -
    We are glad that all reviewer appreciates our work contribution, and we are encouraged by the positive comments on the novelties (Reviewer 3MzD & xTq4& e3Zq & W3XG), adequate experiment settings (All Reviewer), and good paper quality (Reviewer e3Zq & xTq4 & M64c & W3XG).
    Meanwhile, in the individual replies, we have already addressed other comments.

- Future revision
  -
  We have addressed the reviewer's concerns in the point-to-point response. Below are the revisions we will make based on the reviewer's feedback.
  - Additional baselines comparison

    According to reviewers M64c and W3XG, we will add experiments comparing A2PO with the return-conditioned methods DT and %BC, data rebalance methods ReD, policy constraint methods PRDC methods.

  - Additional ablation study

    According to reviewer M64c, we will add an ablation study on A2PO regularization term in actor loss.

  - Modification of the presentation

    According to reviewer xTq4, we will update the description of the policy evaluation paragraph and the corresponding formula. We will correct the incorrect score reports related to baselines in the gym task and update the score reports in the antmaze v0 version environment.

---

### Decision · Program_Chairs · 2024-09-25

**Decision:**

Accept (poster)

**Comment:**

Strong reviews, and all concerns addressed during the discussion period. Clear accept.